# Thinning Antarctic glaciers expose high-altitude nunataks delivering more bioavailable iron to the Southern Ocean

Kate Winter [1] ✉, John Woodward [1], Stuart A. Dunning[2], James R. Jordan [3], Joseph A. Graly[1], Matthew J. Westoby [4], Sian F. Henley [5] & Robert Raiswell[6]

Glacial systems entrain and transfer sediment, rich in essential nutrients, from continental sources to the ocean, where they are released by meltwater. In the Southern Ocean, primary producers are limited by the availability of micro-nutrients, like iron (Fe), so any increase in continental sediment supply could enhance primary productivity and subsequent drawdown of atmospheric $CO_2$. Here we provide a systematic account of labile Fe concentrations in Antarctic continental sediments. Ferrihydrite and crystalline Fe (oxyhydr)oxides were extracted from 27 Antarctic samples collected from nunataks, lateral moraines and blue ice areas in the Sør Rondane Mountains, East Antarctica. We report ascorbate extractable Fe (FeA) in all samples and enhanced precipitation of dithionite extractable Fe (FeD) in subaerially exposed mountain sediments. Our results suggest that as temperatures rise and Antarctic glaciers thin, newly exposed rock surfaces could supply more bioavailable iron to glacier systems, and subsequently the Southern Ocean.

Primary production in the Southern Ocean represents 5–10% of global ocean productivity[1]. Phytoplankton growth and its regulatory effect on climate via atmospheric $CO_2$ uptake are restricted by iron (Fe) supply[2]. Over glacial-interglacial cycles, iron supply to the Southern Ocean from Antarctica has varied significantly and has driven major changes in primary production and carbon export[3–5]. Iron supply to the Southern Ocean has been documented via atmospheric, oceanic, cryogenic and terrestrial mechanisms (e.g. Gerringa et al.[6]; Henley et al.[7]; Tagliabue et al.[8]), with many of these source terms originating from the Antarctic continent and continental shelves[9–11]. Among these, sediment delivered to the oceans via the growth and decay of glacial systems could be a rich source of reactive and bioavailable iron (BioFe) occurring as dissolved, colloidal/nanoparticulate and sediment-bound particulates[12,13]. Previous studies have quantified the potential BioFe content of sediment particles extracted from sea ice[14,15] and iceberg samples[16–18], some coastal sediments[19,20] and atmospheric dust samples[21,22], and in terrestrial streams[23]. However, the abundance of labile iron (as a pre-cursor to BioFe) in Antarctic continental sediments is poorly known, and their delivery routes to the Southern Ocean have not been assessed.

Here we analyse the iron content of sediment collected on and around the 13 km wide Gunnestad Glacier which flows through the coastal margin Sør Rondane Mountains in Dronning Maud Land, East Antarctica (71.94° S, 23.34° E) (Fig. 1). The mountains, which comprise Late Proterozoic to Paleozoic gneiss and amphibolite, sporadically punctuated by km-size plutonic intrusions of granite, syenite and diorite[24], cover an area of ~200 km (latitudinal) by 90 km (long-itudinal), with peaks protruding up to 1.6 km above the East Antarctic Ice Sheet (EAIS). The mountain range includes several nunataks – subaerially exposed mountains surrounded by ice, which punctuate our study site. Whilst large ice streams, like the Western Ragnhild Glacier and Hansen Glacier channelise ice from the polar plateau around the mountain range at speeds >150 m a$^{-1}$, other, smaller ice streams, like Gunnestad Glacier cut through the massif itself, where ice

[1]School of Geography and Natural Sciences, Faculty of Science and Environment, Northumbria University, Newcastle-upon-Tyne, UK. [2]School of Geography, Politics and Sociology, Newcastle University, Newcastle-upon-Tyne, UK. [3]Department of Geography, Faculty of Science and Engineering, Swansea University, Swansea, UK. [4]School of Geography, Earth and Environmental Sciences, University of Plymouth, Plymouth, UK. [5]School of GeoSciences, University of Edinburgh, Edinburgh, UK. [6]School of Earth and Environment, Faculty of Environment, University of Leeds, Leeds, UK. ✉ e-mail: k.winter@northumbria.ac.uk

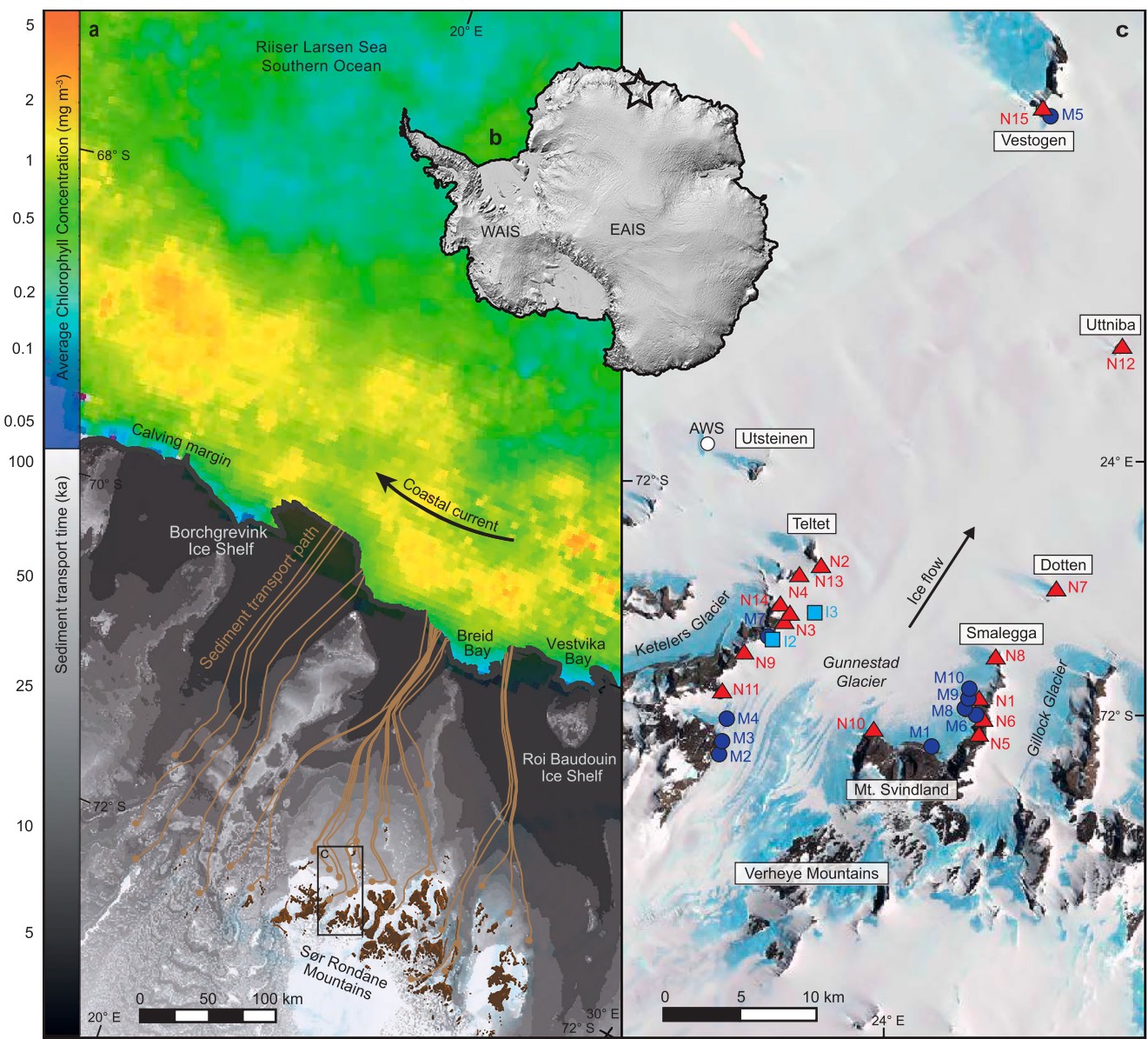

**Fig. 1 | Location Map, showing sample sites, potential sediment tracks and satellite-derived chlorophyll concentrations in the Southern Ocean. a)** Sør Rondane Mountain study site, Dronning Maud Land, East Antarctica (marked with a star on the Reference Elevation Model of Antarctica[69] inset map in **b**, which highlights the West Antarctic Ice Sheet (WAIS) and East Antarctic Ice Sheet (EAIS)). Sediment tracking experiments show how clasts (brown circles) sourced from nunataks or subglacial sources might advect down flow towards the calving margin (along brown lines). The greyscale background image shows first order approximations of the lower bound estimate of sediment transport time to the ice front. The inset box shows the location of panel c. Chlorophyll concentrations in the Riiser Larsen Sea[65] (monthly climatology for January over the period 2003–2025) show that phytoplankton blooms along the ice front - where sediments are deposited by melting ice and transported west, along the continent, by local currents. **c** Landsat Image Mosaic of Antarctica[70] showing sediment sample collection sites from nunataks (red triangles), moraines (blue circles) and the ice surface (light blue squares). Text boxes highlight named mountain ranges and nunataks. All maps were produced using the Quantarctica mapping environment[68] in QGIS.

flows at ~10–30 m a$^{-1}$ over the mountainous subglacial terrain[25]. Cosmogenic nuclide dating and geomorphological analysis of local tills, boulders and trimlines suggest a gradual lowering of the ice surface of at least 500 m since the Early Pleistocene[26] with only limited, localised evidence for reburial[27]. This means that subaerially exposed bedrock has had ample time to weather and break down[28], generating an important source of loose, available sediment close to the glacier surface. Mass movement processes in the mountains deliver these sediments to glaciers below, where subsequent snow fall and ice flow bury and transport this subaerially sourced sediment, as well as sediments sourced at the ice-bed interface through glacier scour and plucking, towards the coast[29].

Our simulated sediment transport routes in Fig. 1a show how sediments congregate down-flow, towards the centre of ice flow units, focusing sediment delivery along discrete coastal outlets. Most glacially scoured, basally transported sediment will be released by melting ice close to the grounding line, and within large grounding zone wedges[30,31]. For iron contained within these sediments to reach the well-lit surface ocean where it can be utilised by phytoplankton, it must be resuspended or dissolved by currents flowing under the ice shelves and transported beyond the ice shelf front[32,33]. As such currents flow under the ice shelves, they can melt the underside of the ice, entraining more glacially-sourced iron, and mix with meltwater which increases their buoyancy[34,35]. The extent to which this iron reaches surface-ocean

phytoplankton beyond the ice shelf front is controlled by the degree of buoyancy modification due to ice-ocean interactions and subsequent mixing of the iron-enriched water mass by wind, sea ice and iceberg-stirring processes[36,37]. In contrast, sediments entrained higher up the ice column, or at the ice surface (delivered by nunatak weathering) are more likely to remain in high level englacial transport[38], where ice and sediment can be transported by glacier flow, past the grounding line, to the calving margin. Here, high basal melt rates and iceberg calving[39] release sediment-derived iron to the adjacent coastal ocean region via coastal currents, either directly or after transport by free-floating icebergs and subsequent release upon ice-melting and resultant mixing of the water column[40]. The biological effect of these subglacial/basally-transported and englacial or supraglacial iron sources is challenging to constrain directly, but they are likely to contribute iron supply to the large phytoplankton blooms that occur along the coastline[41] (Fig. 1a), which drive a seasonal $CO_2$ sink in this region[7]. In this study, we examine the inland source of these sedimentary iron-supply mechanisms and quantify the iron concentrations of sediments derived from subaerial and subglacial sources, to assess the impact of thinning, melting ice on iron delivery to the Southern Ocean.

## Results

Here we analyse 27 samples collected along nunatak margins, lateral moraine accumulations, in wind-scooped surface depressions on the ice, and within glacial sediment bands where glacially sourced sediments are transported to the ice surface in wind-scoured blue ice areas[42]. These settings capture a variety of environments in Antarctica, where surface sediments accumulate as a result of wind, blue ice flow, glacial flow and mass-movement processes in subaerially exposed mountain settings (which dominate large swathes of coastal and near-coastal Antarctica). Chemical sediment extractions for amorphous and nanoparticulate ferrihydrite and crystalline Fe (oxyhydr)oxides were carried out according to methodology published by Raiswell et al.[43] (see Methods). For each sample, we established the wt. % Fe that is extractable by ascorbic acid (FeA) – as FeA has been shown experimentally to be bioavailable for phytoplankton[17,44,45] and favourable for microbial reduction[46,47]. We also employed a dithionite (FeD) treatment, which acts to dissolve (or at least partially dissolve) residual minerals such as aged ferrihydrite, schwertmannite, lepidocrocite, goethite and hematite[48]. Whilst we note that at the point of delivery to seawater, ferrihydrite (measured as FeA) is more labile than FeD, previous study has shown that FeD may become more bioavailable after delivery to seawater, due to processes that include the in-situ recycling of cellular iron through grazing and viral lysis[49]. As a result, research suggests that Southern Ocean productivity is enriched by both FeA and FeD.

Table 1 shows sediment extraction results for FeA and FeD (as well as the ratio FeD: FeA) and total Fe (FeT), determined by X-ray fluorescence (XRF) spectrometry (see Methods and Supplementary Data 1), for samples broadly grouped into three categories: nunatak samples (collected close to the ice surface), moraine samples (from lateral moraines and blue ice sediment bands) and wind-blown sediments collected on the ice surface (sample sites noted in Fig. 1c). We report concentrations of labile Fe in all sediment samples. FeA concentrations are quite similar across our study region with average weight percentages of $0.016 \pm 0.010$ in nunatak samples, $0.019 \pm 0.007$ in moraine samples and $0.012 \pm 0.002$ in ice surface samples, and are comparable to FeA values (0.015) from samples collected from Greenlandic icebergs and melting ice[50]. In contrast, FeD concentrations vary depending on collection site, with nunatak samples having the highest average FeD wt. % of $0.276 \pm 0.319$, which is over three times greater than both the moraine samples ($0.090 \pm 0.025$) and the ice surface samples ($0.069 \pm 0.028$). The highest FeD: FeA ratios of 39.8 and 28.6 are reported in samples N3 and N4 (Fig. 2), collected along nunataks that define the lateral margins of Gunnestad Glacier (Fig. 1c). These

visibly iron-stained sediment samples (see photograph in Supplementary Fig. 1a) have FeD percentages of $0.989 \pm 0.181$ and $1.000 \pm 0.228$, which are up to an order of magnitude greater than any other reported FeD data from the Antarctic continent[17]. Sediments with no visible iron staining have much lower FeD concentrations of between $0.045 \pm 0.01$ and $0.231 \pm 0.065$ wt. % (similar to FeD extractions from sediments collected in West Antarctica which report 0.027 to 0.45 wt. % (n = 8)[17]). Our results show appreciable FeA and FeD across our study region, and significant amounts of FeD (particularly in visibly iron-stained bedrock) in exposed mountain areas; more work is now required to understand how FeA and FeD varies among different glacial systems across the continent.

FeT concentrations also vary across our study region (although this could not be analysed on ice surface samples, due to a lack of material). Overall, FeT ranges from 1.661 to 10.558 wt. %, with an average of 5.849 wt. %. Despite the greatest FeT concentrations being recorded in two of the iron-stained nunatak samples (10.353 and 10.558 wt. %), differences in FeT between moraine samples (average: $6.423 \pm 1.955$) and nunatak samples (average: $5.480 \pm 2.822$) are not statistically significant (Table 1; p = 0.36). This implies that the higher FeD in the nunatak samples is not due to differences in composition but due to weathering processes occurring on the nunatak surface. As such, enhanced FeD may be a feature of similarly exposed nunataks elsewhere in Antarctica, even when lithology and particle make-up differ.

We further explore Antarctic sediment weathering processes with scanning electron microscopy (SEM) and elemental analysis (see Methods). Loose precipitates and accretion nodules in nunatak samples (like N6; Fig. 2b, c) suggest that high FeD values might derive from mineral transformation and precipitation, assisted by the conversion of ferrihydrite (-FeA) to goethite/haematite (-FeD), which has a half-life of around 4-5 years at pH 7 and 5 °C (in laboratory simulations)[51]. Whilst this transformation process would be slower in Antarctica, because air temperatures rarely exceed 0 °C, there is ample time for the transition to occur, and the process will speed up in the austral summer when rock temperatures soar, as a result of solar gain (see Supplementary Fig. 2 and Supplementary Data 2). Although no accretion nodules are recorded in our moraine samples, grain coatings and loose precipitates (Fig. 2d, e) show that precipitation of especially amorphous or poorly crystalline phases of iron may occur in the glacial/subglacial environment, as recorded elsewhere in Antarctica[52,53]. Our results highlight the importance of weathering in low temperature systems, within and potentially beyond Antarctica.

## Discussion

The abundance of FeA and FeD in nunatak and glacially derived/transported sediments across our glaciated mountain study site, combined with similar, reported labile (and potential BioFe) concentrations offshore, in icebergs[17], show that FeA and FeD can be glacially transported from their mountain source to the ocean. We use present day ice surface velocities to simulate the potential flow paths and transport time of nutrient delivery from the Sør Rondane Mountains to the coast (see Methods) to explore where labile iron may be deposited. The resultant map, in Fig. 1a, suggests, as a simple first order approximation of time, that it will take at least 10–100 ka for sediments sourced in the mountains to reach the ice front: highlighting a significant lag between any changes in nutrient supply and subsequent Southern Ocean delivery. These lag times are likely applicable across much of Dronning Maud Land, where coastal margin mountain range sediments are picked up by ice flows and transported similar distances to the coast. It is hard to quantify how material will change during transport, but we expect that dissolution and reprecipitation may occur as re-freezing concentrates iron, and that there may be particle breakdown during these processes, which may act to increase iron bioavailability.

**Table 1 | Composition of sediment samples collected from 3 different environments: nunataks (N), glacial moraines (M) and the ice surface (I)**

|  | Sample | FeA wt. % | FeD wt. % | FeD: FeA | FeT wt. % |
|---|---|---|---|---|---|
| Nunatak Samples | N1* | 0.009 | 0.140 | 14.894 | 4.766 |
|  | N2* | 0.039 | 0.572 | 14.774 | 3.058 |
|  | N3* | 0.025 ± 0.005 | 0.989 ± 0.181 | 39.787 | 10.558 |
|  | N4* | 0.035 ± 0.003 | 1.000 ± 0.228 | 28.607 | 10.353 |
|  | N5 | 0.009 ± 0.002 | 0.082 ± 0.031 | 8.708 | 5.252 |
|  | N6† | 0.021 ± 0.004 | 0.231 ± 0.065 | 11.051 | 4.418 |
|  | N7 | 0.003 ± 0.001 | 0.067 ± 0.012 | 22.501 | 1.742 |
|  | N8 | 0.010 ± 0.003 | 0.105 | 10.853 | 3.843 |
|  | N9 | 0.018 | 0.249 | 13.887 | NA |
|  | N10 | 0.013 ± 0.001 | 0.152 ± 0.085 | 12.061 | 7.956 |
|  | N11 | 0.019 ± 0.002 | 0.083 ± 0.014 | 4.372 | 6.214 |
|  | N12 | 0.022 ± 0.002 | 0.133 ± 0.021 | 6.188 | 5.197 |
|  | N13 | 0.006 ± 0.000 | 0.059 | 9.811 | 1.661 |
|  | N14 | 0.009 ± 0.001 | 0.220 | 25.575 | 3.819 |
|  | N15 | 0.010 | 0.061 | 6.321 | 7.885 |
|  | Average | 0.016 ± 0.010 | 0.276 ± 0.319 | 15.293 ± 10.108 | 5.48 ± 2.822 |
| Moraine Samples | M1 | 0.019 | 0.133 | 7.035 | 7.651 |
|  | M2 | 0.031 ± 0.004 | 0.092 ± 0.023 | 2.952 | 4.581 |
|  | M3 | 0.022 ± 0.004 | 0.096 ± 0.027 | 4.406 | 8.935 |
|  | M4† | 0.015 ± 0.003 | 0.045 ± 0.01 | 3.078 | 7.075 |
|  | M5 | 0.007 ± 0.001 | 0.058 ± 0.017 | 7.719 | 3.49 |
|  | M6 | 0.021 ± 0.002 | 0.101 ± 0.023 | 4.841 | 3.826 |
|  | M7 | 0.016 ± 0.004 | 0.087 ± 0.030 | 5.494 | NA |
|  | M8† | 0.026 ± 0.007 | 0.082 ± 0.018 | 3.126 | 7.453 |
|  | M9 | 0.021 ± 0.006 | 0.103 ± 0.018 | 4.822 | 7.956 |
|  | M10 | 0.009 ± 0.000 | 0.100 ± 0.004 | 11.406 | 6.843 |
|  | Average | 0.019 ± 0.007 | 0.090 ± 0.025 | 5.488 ± 2.630 | 6.423 ± 1.955 |
| Ice Surface | I1 | 0.011 | 0.049 | 4.522 | NA |
|  | I2 | 0.014 ± 0.009 | 0.088 | 6.524 | NA |
|  | Average | 0.012 ± 0.002 | 0.069 ± 0.028 | 5.523 ± 1.415 | NA |
| T-Tests (p value) | Nunatak v. Moraine | 0.5208 | 0.0402 | 0.0019 | 0.3550 |
|  | Nunatak v. Surface | 0.1874 | 0.0269 | 0.0030 | NA |
|  | Moraine v. Surface | 0.0410 | 0.4656 | 0.9802 | NA |

FeA samples (representing the wt % extractable by ascorbic acid) and dithionite treated FeD samples were analysed in full process triplicate where possible, so results here are reported as averages with standard deviations of the sample noted by ± values; those without ± value were processed only once. Samples with * displayed visible signs of iron staining. Samples with a 'underwent SEM analysis. FeD: FeA is a unit-less ratio. NA = sample not analysed by XRF for FeT (due to a lack of sample).

With no evidence for tributary flow switching or shut down of Gunnestad Glacier, and only a modest lowering of the ice surface at our study site since the Early Pliocene, we expect that continentally derived, potentially BioFe rich sediment will have been transported along similar transport routes for hundreds of thousands of years, at least. Provided long term sediment storage is avoided (e.g. thick moraine accumulations in nunatak embayments[54]), delivery to the ocean will be inevitable, and rates of subaerially-sourced, glacially transported sediment entering the ocean will likely increase as glaciers thin in a warming world.

The potential for iron delivery to enhance productivity in the ocean is supported by elevated chlorophyll concentrations (a proxy for phytoplankton biomass) in surface waters adjacent to, and offshore from, the ice flows supplied by Gunnestad Glacier and other neighbouring glaciers (Fig. 1a). Increases in phytoplankton growth resulting from glacial iron sources are well documented around Antarctica (e.g. Gerringa et al.[6]; Alderkamp et al.[55]; Herraiz-Borreguero et al.[34]). A study of 46 Antarctic coastal polynyas including Breid Bay and Vestvika Bay, which abut Borchgrevink and Roi Baudouin ice shelves respectively,

shows that variation in phytoplankton biomass among polynyas is explained mostly by ice shelf melt rates (and inferred iron supply) and secondarily by iron upwelled from sediments[41]. Here, satellite-derived chlorophyll *a* concentration for January (when peak austral summer net primary production occurs), averaged over ~20 years shows annually recurrent phytoplankton blooms along the coastline, extending up to 175 km from the ice margin, which is well within the range of floating, melting icebergs[34]. These findings support the previously documented biological importance of glacially transported nutrient rich sediment delivery to the Southern Ocean.

As temperatures rise and ice sheets thin, more bedrock will be exposed in glacial environments, increasing both the contributing area for mass movement processes to deliver more sediment to glacial systems and, as we show in our work: the concentration and ratio of iron phases, through the increased abundance of fresh FeA and weathered FeD. Indeed, it is this this transient low temperature surface weathering resulting in nunatak alteration that makes Antarctica a potential analogue for many other sites, including Martian weathering, where comparable surface temperature ranges are recorded[56]. We

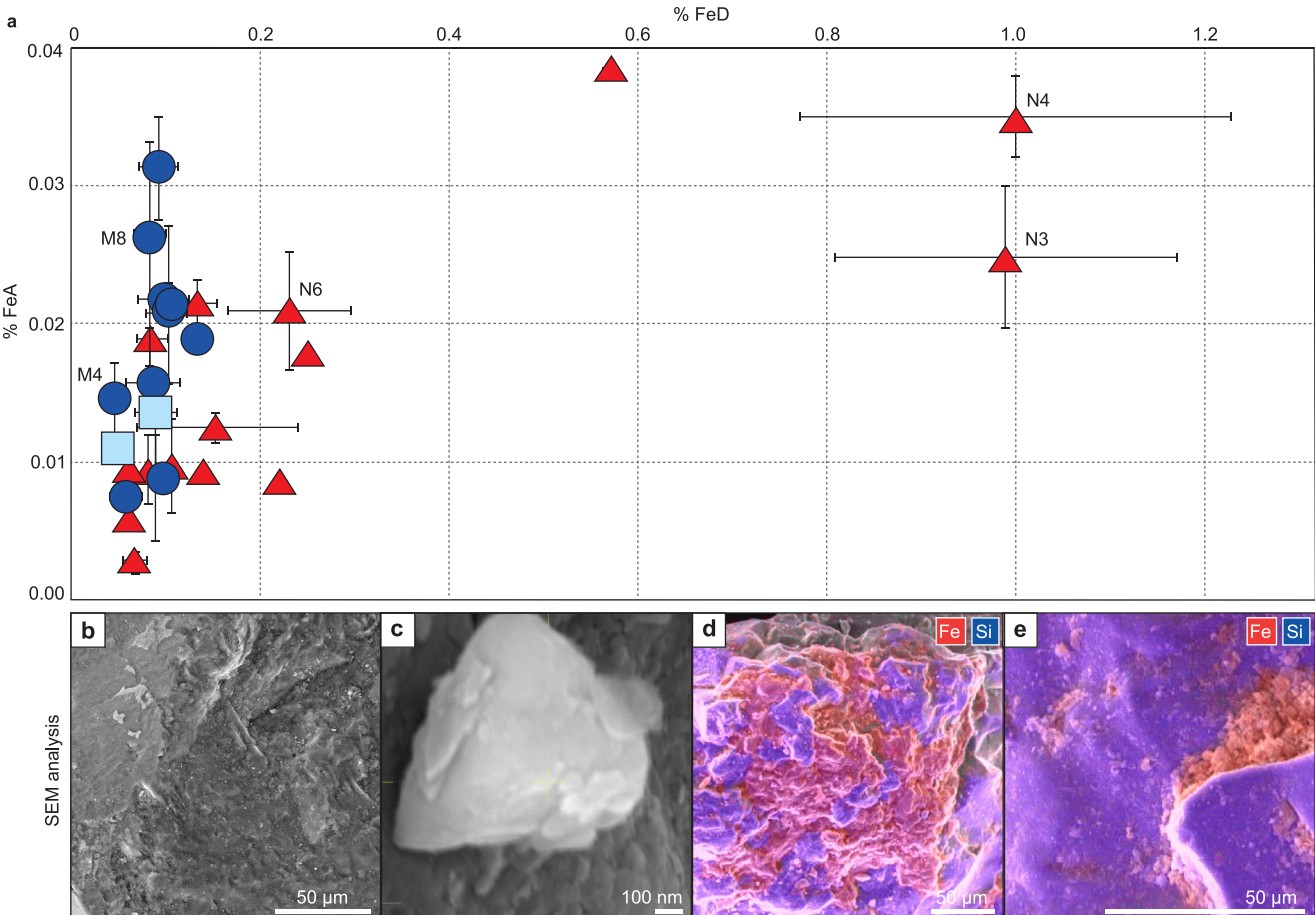

**Fig. 2 | Geochemical analysis of 27 Antarctic sediments. a**) a graph showing the relationship between iron that is extractable by ascorbic acid (FeA) against iron that is extractable by dithionite solution (FeD), where different collection sites are represented by symbology/colouration (red triangles for nunatak samples, blue circles for moraine samples and light blue squares for ice surface samples). Error bars show standard deviations (when analysis was conducted in triplicate) and labelled samples are individually discussed in the text. Panels b-e show results from scanning electron microscope (SEM) and elemental analysis of Antarctic sediments. Panels **b** and **c** showcase weathered on iron nodules, clustered around surface cracks on nunatak sample N6. Elemental analysis of glacial moraine samples reveal iron coating of quartz grains (rich in silica (Si)), for example in sample M4 (**d**) and loose, bio-available iron accumulations in depressions, shown by red colouration, in sample M8 (**e**).

postulate that melt/freeze processes at glacial ice/bed interfaces in Antarctica may increase sediment supply below the ice surface, as warmer, wetter based ice systems may more easily dislodge and entrain sediments[57]. With all other factors being consistent, we hypothesise that these increases in sediment availability, and transfer through glacial systems, will allow more iron to be transported to the adjacent coastal ocean, increasing the potential for phytoplankton growth, with resultant implications for the drawdown of atmospheric $CO_2$. Whilst the effects of climate warming may also accelerate the transfer of this sediment-derived iron from its mountain source to the ocean[29], there is still likely to be a significant time lag between changes in sediment supply to glacial systems from nunataks and changes in iron supply to the ocean (Fig. 1). As such, the mechanisms proposed here are likely to integrate processes over much longer timescales than other iron supply mechanisms to the Southern Ocean (e.g. Henley et al.[7]).

Our hypothesis for changing iron delivery from Antarctic mountains to the Southern Ocean is explored in Fig. 3. Figure 3a depicts a thick ice sheet where the underlying mountains are covered in ice – like the area to the west of our study site (see Fig. 1a), which is representative of much of East Antarctica. In this area, there are no subaerially exposed sediment sources, depriving the glacier sediment conveyor system of FeD-rich weathering and mineralisation products. However, FeA products (and some FeD) will still be

picked up at the glacial bed through glacial scour, where clasts will mainly be transported as basal load, though higher level transport may occasionally occur when glacial thermal conditions allow[42]. As most sediment in this environment is transported close to the bed, it will be deposited as till continuum close to the grounding line and beneath floating ice. From there, it must then be resuspended by under-ice currents and transported beyond the ice shelf front and mixed into the well-lit surface ocean in order for its iron to be directly useful to phytoplankton. As such, only a portion of the glacially-transported sediment-derived iron is available to fuel phytoplankton photosynthesis and carbon uptake in the Southern Ocean.

Figure 3b represents the current situation in the Sør Rondane Mountains (and much of Dronning Maud Land) where the ice sheet close to the margin is thinner than the deep interior ice, allowing nunataks to deliver FeA and FeD rich sediments to ice flows below through weathering, mass movement and burial processes. As these glaciers continue to entrain iron rich subglacial sediment too, iron will be transported coastwards at all elevations within the ice column. This permits sediment delivery to the grounding line and the base of floating ice shelves, from which under-ice currents can entrain sediment-derived iron and transport it to well-lit surface waters, as well as the calving margin, where sediment-derived iron is released to the coastal ocean either directly or after iceberg

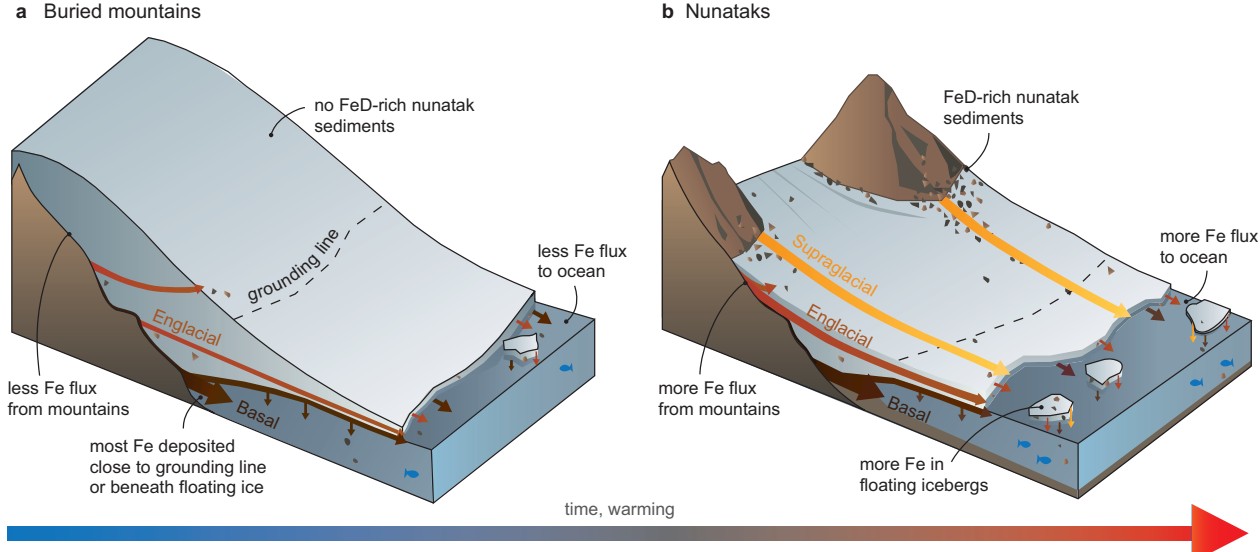

**Fig. 3 | Schematic diagram depicting sediment and iron (Fe) transfer through Antarctic glacial margins.** Panel **a** represents a situation where there are no exposed mountains (like the area to the west of our study site, and a large proportion of Antarctica), so there are no surface sediments rich in dithionite-extractable Fe (FeD). Sediments can only be sourced from the ice/bed interface when glacial thermal conditions permit entrainment. While most sediment will be transported along the bed, as basal load, englacial transport may occur (if glacial conditions permit entrainment into the ice). Panel **b** represents an ice flow through exposed coastal margin mountains (analogous to much of Dronning Maud Land, including our subaerially exposed study site). Here, sediments derived from weathering and mass movement processes in the mountains are either transported supraglacially or incorporated in ice flow through burial by subsequent snow/rock fall. Just like panel a, sediments are also transported englacially, and along the glacier bed. Whilst both scenarios deliver nutrients to the Southern Ocean, thick ice sheets flowing over buried mountains will mainly transport sediments as basal load, so deposition will be released as subsurface melt close to the grounding line, where light is more limited for photosynthesis (and resultant $CO_2$ drawdown). However, where ice is thinner and bedrock is exposed, more sediments (rich in both FeA and FeD) will be available for entrainment, where a larger percentage of supra-and englacial transport will deliver more nutrient-rich sediment to the open ocean, where it can melt and rain out of free-floating icebergs.

transport and melting. Transport of iron-rich sediment in icebergs and subsequent release may be particularly important for distributing potentially bioavailable iron over larger areas of the coastal ocean[58], and even beyond the shelf break to the open Southern Ocean[39,40]. Similar to much of coastal Antarctica, the westward-flowing Antarctic Coastal Current[59] interacts strongly with currents emerging from ice shelf cavities in our study region[60] (Fig. 1). This interaction likely facilitates the transport of iron-enriched waters and melting icebergs downstream, thus further increasing the area over which Southern Ocean productivity benefits from continentally derived, nutrient-rich Antarctic sediments.

As the EAIS thins in response to climate change[61], we expect that areas of Antarctica that are currently covered by ice and therefore nunatak free will become more like our study region, where nunataks are exposed above the ice surface, increasing the potential contributing area of subaerial sediment supply to Antarctic ice. As air temperatures continue to rise, thermo-mechanical changes, including those induced by permafrost degradation, will increase the magnitude and frequency of de-buttressed rock slope failures (as recorded in other cold regions)[62,63], contributing more sediment to glacial systems. We hypothesise that these factors will increase sediment (and iron) flux from subaerial sources. High-level glacial transport will help to deliver these newly exposed/released nunatak sediments beyond the grounding line to calving margins, where meltwater and free-floating icebergs can release sediment-derived nutrients over large areas of the coastal ocean, where they can be utilised by primary producers. Our work highlights how enhanced high-altitude weathering, increasing sediment contributing area and efficient sediment delivery could impact iron flux from the Antarctic continent, influencing the strength of the Southern Ocean biological carbon pump and atmospheric $CO_2$ drawdown over long timescales.

## Methods

### Sediment collection

Antarctic sediments were collected from nunataks, moraines, the glacial surface and emergent englacial debris bands in wind scoured blue ice areas in and around Gunnestad Glacier, East Antarctica (Fig. 1c). Nunatak samples, comprising of in-situ loosened, weathered bedrock were collected close to the glacier margin, no more than 10 m above the ice sheet surface. Whilst bedrock at this elevation will have been entombed in ice in the past, local cosmogenic nuclide studies suggest that these sampling sites have been aerially exposed and ice-free for some thousand years at least (based on a surface lowering rate of ~10 m every 3000 years since thinning from the Last Glacial Maximum started in this area (around 14 ka)[26]. While there are no visible signs of water or water pooling at our nunatak sites, high bedrock surface temperatures (Supplementary Fig. 2) quickly melt away any snowfall or snowdrift in summer (particularly on north facing slopes). This snowmelt, combined with significant diurnal surface temperature contrasts, generates the liquid water necessary for weathering and bedrock erosion (e.g. frost-shattering), as well as the alteration of minerals through weathering processes.

Moraine samples resting on glacial ice were collected along lateral moraines at the glacial margin and across medial moraines, which form at the interface of different ice flow units, as well as within debris bands exhumed by surface ablation-controlled ice flow in blue ice areas[42]. In all cases, clasts are lithologically varied and poorly sorted, reflecting a variety of sources and glacial transportation mechanisms (both on the surface, and through the glacier). Whilst some moraine sediments (usually ones deep within lateral moraine assemblages) show visible signs of mineral precipitation, like iron staining, other, fresher looking clasts (sometimes partially entrained in glacial ice) suggest more recent glacial transportation. We focus our data collection and analysis on these partially entrained moraine clasts and sediments as they are

the most available for glacial transportation (and therefore nutrient delivery to the coast), as sediments deep within extensive lateral moraine embayments are unlikely to be transported to the coast without significant changes in glacial flow[54].

Two other samples were collected in wind-scoured hollows on the ice surface, which may represent incipient cryoconite traps[64]. Both sites contained a small number of sediment grains (>1 mm in size). These sediment collection sites are quite rare in our study region and we expect that they represent wind-blown material sourced from further up-glacier - as wind at our study sites generally transcends from the polar plateau, towards the Southern Ocean.

All sediments were collected during the austral summers of 2018–2019 and 2019–2020. Samples were photographed prior to collection (e.g. Supplementary Fig. 1), where loose sediments or clasts were scooped into a plastic vial, sometimes with the aid of a plastic spatula. Site locations were recorded on a handheld Garmin GPS and samples were stored in a cool box on the back of a snowmobile (for no more than 3 h), before they were transferred to a freezer (consistently below −15 °C) at the Princess Elisabeth Antarctica Research Station. All sediments remained frozen prior to laboratory analysis.

## Chemical extractions and sediment analysis

Chemical sediment extractions for amorphous and nanoparticulate ferrihydrite and crystalline Fe (oxyhydr)oxides were carried out according to methodology published by Raiswell et al.[43]. For each sample, we established the wt. % Fe that is extractable by ascorbic acid (FeA) – as FeA has been shown experimentally to be bioavailable for phytoplankton[17,44,45] and favourable for microbial reduction[46,47]. We also employed a dithionite (FeD) treatment, which acts to dissolve (or at least partially dissolve) residual minerals like aged ferrihydrite, schwertmannite, lepidocrocite, goethite, and hematite[48]; all of which are relatively stable, but possibly less bioavailable[17]. Whilst we note that at the point of delivery to seawater, ferrihydrite (measured as FeA) is more labile than FeD, research has shown that FeD may become more bioavailable after delivery to seawater, from the in-situ recycling of cellular iron through grazing and viral lysis[49].

All sediment samples were air dried in a fume cupboard at room temperature for 1 week prior to chemical extraction. Each loose sediment sample was sieved through an acetone-cleaned 1 mm nylon mesh sieve to remove coarse material. Filtered solutions of ascorbic acid (containing 200 ml of deoxygenated distilled water, 10 g tri-sodium citrate and 10 g sodium bicarbonate, made up to pH 7.5 with ~10 ml ascorbic acid) and dithionite solution (containing 50 g sodium dithionite, 0.35 M acetic acid and 0. 2 M sodium citrate (58.82 g/L), adjusted to pH 4.8 using hydrochloric acid) were added to each sample separately. The extractions were performed over a 24 hr period, shaking 30–40 mg of sample and 10 ml of extractant at room temperature. Where sample size permitted, the analyses were performed in triplicate, with 3 full process replicates, allowing assessment of reproducibility. FeD is calculated as the measured FeD less the measured FeA. Solutions were analysed using Northumbria University's Inductively coupled plasma - optical emission spectrometer (ICP-OES) Perkin-ELMER OPTIMA 8000 (where blanks were sub 10 ppb, and triple replicates of measured values showed a mean standard deviation of ±1%).

X-ray fluorescence (XRF) spectrometry, which records the elemental composition of finely ground sediment samples was employed to examine major element concentrations, including total iron (FeT). Homogenous pellets were created by fusing finely ground sediment samples with a Cereox BM-0002-1 (Fluxana, Germany) wax binder (shaken together for 15 s), using a 10:1 sediment to binder ratio. These pellets (and wax pellets used as procedural blanks) were analysed using Northumbria University's Spectro XEPOS XEP05 ED XRF benchtop analyser.

A TESCAN MIRA3 Field Emission Gun - Scanning Electron Microscopy (SEM) was employed to analyse sediments from different environments on a micro and nano scale. Samples from nunatak and moraine locations were defrosted at room temperature over a 48-h period then freeze dried using liquid nitrogen and left in the freeze drier for 18 h. These sediments were then pressed onto a SEM mount using conductive carbon adhesive tape before nitrogen was blown over each mount in turn, to remove any loose sediment particles. All SEM mounts were coated with platinum (5 nm thick) for preservation prior to SEM analysis – which included elemental analysis using an Oxford Instruments X-max energy-dispersive X-ray spectroscopy (EDS) beam.

## Particle tracking experiments

To determine both the flow path and approximate time it takes for sediment particles sourced from inland rocky-outcrops to deposit out of the ice sheet we performed simple particle tracking experiments in MATLAB using the MeASURES ice velocities data set[25] (Fig. 1a). We "released" a particle into the domain at the ice-nunatak boundary and advected it by the surface ice velocity at its current location for a fixed amount of time (for the purpose of this work chosen to be 100 years). This period is long enough to allow sediment to move out of source areas (and dampen the impact of errors in current ice velocity data), but short enough for us to record changes in areas with fast flowing ice. The process of advection over 100 years is repeated from the particles new location until the particle leaves the domain (at the coast), with both the path it travelled, and the time taken to do so, determined. This methodology requires three main assumptions: 1) Ice velocity is constant through time - at its present day observed values. 2) The vertical profile of the ice velocity is the same as the surface velocity. 3) Sediment particles are transported as fast as the ice velocity. These assumptions are necessary due to the lack of direct observations both temporally (we have no records of ice velocity dating back thousands of years) and spatially (we have few direct measurements of how horizontal ice velocity changes within a vertical column), and of the process of sediment transport within the ice column. As such, these results are a first order approximation of potential sediment (and therefore nutrient) flow paths, where transport time to the ice front should be thought of as the lower bound of time taken for sediment deposit, with actual time taken likely to be longer due to subsurface ice velocities possibly being lower than surface velocities and sediment potentially being transported at a rate slower than ice velocity.

## Chlorophyll extractions

Surface chlorophyll concentrations in the Riiser Larsen Sea of the Southern Ocean were extracted from Level 3 MODIS Aqua colour data[65], presented here as the monthly climatology for January over the period 2003–2025. As there are some concerns over the accuracy of chlorophyll concentrations from satellite-derived ocean colour measurements in high-latitude waters, we focus on relative concentrations rather than absolute values. Another concern, that satellites cannot detect chlorophyll below the upper optical depth of the ocean (and therefore cannot measure chlorophyll at depth), is partially alleviated by the very high correlation between surface chlorophyll and depth-integrated chlorophyll within the mixed layer of the Southern Ocean[66]. As a result, the relative concentrations of chlorophyll presented in Fig. 1a are representative of total concentrations at the temporal and spatial scales examined here.

## Nunatak temperatures

All nunataks show visible signs of weathering, where many of the exposed mountains are snow-free and covered in a thick layer of loose sediment, which appears to have shattered and broken down in-situ. We use air and rock temperature measurements to monitor ground conditions and the frost-shattering window. Konrad Steffen (former

Director of the Swiss Federal Institute for Forest, Snow and Landscape Research) provided hourly air temperature data from his custom-built Automatic Weather Station (AWS) maintained close to Utsteinen Nunatak (Fig. 1) by staff at the Princess Elisabeth Antarctica Research Station. The AWS uses a Campbell data logger to record air temperatures detected by a Vaisala HMP45 temperature sensor. Hourly bedrock surface temperature measurements were recorded with a MadgeTech CryoTemp Ultra Low Temperature Data Logger, secured to a dark-coloured bedrock spur (next to the nunatak sediment sample N11, Fig. 1c).

Whilst daily averaged air temperature and rock surface measurements are well below zero during the long austral winter (generally −20 °C to −30 °C), and air temperatures remain low during the austral summer (with temperature records rarely exceeding 0 °C at the local AWS), exposed bedrock is often warmer than 0 °C in summer (see Supplementary Fig. 2). Over the course of ~1 year (1st February 2019 – 20th January 2020) we found that 50% of daily averaged bedrock surface temperature measurements were above 0 °C in the austral summer period. During this time, diurnal fluctuations in rock temperature up to (and occasionally exceeding) 25 °C were common (Supplementary Fig. 2) and in line with observations from other Antarctic sites[67]. A peak rock temperature measurement of 23.8 °C was recorded at midday on the 20th of December 2019 when local air temperature was −6.5 °C. Bedrock temperature dropped to −4.1 °C during that polar night, maintaining a temperature ~7 °C higher than the air temperature recording of −11.4 °C at the same time. Whilst slight variations in air temperature are expected, even across such short travel distances, these elevated rock surface temperatures highlight the importance of solar gain in Antarctica, and the thermal capacity of bedrock. Its these high surface temperatures which keep nunataks snow-free in summer and provide the liquid water necessary for weathering and chemical alteration of bedrock and sediments. These weathering conditions in cold, arid environments are useful analogues for present day weathering on Mars.

## Data availability
Satellite imagery and surface flow speed data presented and used in Fig. 1 are freely available to download from Quantarctica[68] (using specific data references therein): https://www.npolar.no/quantarctica/. MODIS-Aqua Chlorophyll Concentration data[65] used in Fig. 1 are available at: https://oceancolor.gsfc.nasa.gov/l3/. Geochemical data generated in this study are available at https://doi.org/10.25405/data.ncl.30306862.v1. Rock/air temperature data are available at https://doi.org/10.25405/data.ncl.30306883.v1.

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

## Acknowledgements

Antarctic fieldwork was funded by the Baillet Latour Antarctica Fellowship, awarded to K.W. in October 2018. The Polar Regions Department (UK) provided K.W. with a permit for rock collection in Antarctica (permit no. 24/2018). J.R.J. received funding from PROTECT, a European Union's Horizon 2020 research and innovation program, under grant agreement No 869304. We are grateful for the help of all International Polar Foundation staff working at the Princess Elisabeth Antarctica research station and would like to note that data collection would not have been possible without the expert guidance of our field guide, and station doctor, Jacque Richon, as well as our enthusiastic field assistants James Linighan and Ross Winter. The authors are grateful for local air temperature data provided to them by Konrad Steffen (former Director of the Swiss Federal Institute for Forest, Snow and Landscape Research) who set up the AWS near Princess Elisabeth Antarctica Research Station. We thank technicians working within the Geography Department at Northumbria University for their assistance with sediment analyses and are appreciative of the constant development of free and accessible datasets and software packages used in this study: NASA Ocean Biology Processing Group MODIS-Aqua data[65], the MeASURES ice velocities data set[25], The Reference Elevation Model of Antarctica[69], The Landsat Image Mosaic of Antarctica[70] and Quantarctica[68].

## Author contributions

K.W., J.W., S.A.D. and M.J.W. conceived the project and K.W. carried out the fieldwork. Geochemical analysis was conducted by K.W., J.A.G. and R.R. whilst J.R.J. conducted flow-line simulations. Figures were created collaboratively by K.W., J.W., J.R.J, S.F.H and J.A.G. K.W. led the writing of the paper, with all authors contributing to and editing the text.

## Competing interests

The authors declare no competing interests.
