## [Peer Review file · Nature Communications]

Thinning Antarctic glaciers expose high-altitude nunataks delivering more bioavailable iron to the Southern Ocean

Corresponding Author: Dr Kate Winter

Version 0:

Reviewer comments:

Reviewer #1

(Remarks to the Author)

Thinning Antarctic glaciers expose high-altitude nunataks, delivering more bioavailable iron to the Southern Ocean. Winter et al., present measurements of iron (Fe) lability from exposed mountain ridges in Antarctica. By modeling the route of sediment entrained in ice and carried to the coastline some time later they argue that the sediment has a fertilization effect increasing chlorophyll distribution in the adjacent Southern Ocean. The leaching techniques used to establish Fe lability are well established as useful and practical methods to roughly constrain the more-reactive Fe phases of most relevance to short-term biogeochemical cycles, rather than harder and more crystalline phases which are less reactive. The techniques as applied herein are consistent with prior work in other polar regions, which were largely pioneered by Raiswell et al.. For obvious logistical reasons, a notable deficiency of prior work is that much of the sedimentary analysis to date has been conducted on samples from other glaciated regions in Greenland, Svalbard, and other lower latitude regions, with Antarctica poorly sampled by comparison. A strength of this new work is that it is based in an under sampled region.

The main weakness of the work is that the narrative and links to oceanographic processes are more challenging to address and I think these aspects of the work are quite weak and not so well supported as the core geochemistry. The main hypothesis is that labile Fe in sediment moving along ice flows from nunataks has a downstream fertilizing effect in the ocean, but -whilst plausible- demonstrating this with modern day data is difficult as presented. First, the timescales are so long (10-100 ka) that there would be substantial de-coupling between any processes occurring at the nunataks and at the calving ice front or basal ice shelf at the coastline. Second, the basic chlorophyll images used are not proof of a fertilizing effect e.g. chlorophyll is generally high in all coastal regions (globally and more specifically around Antarctica) irrespective of what nutrient/light/grazing limitation patterns are present in the coastal zones, and whether or not there are glacier/iron fluxes at the coastline. These levels of elevated chlorophyll, for obvious reasons, follow the prevailing currents which are likely alongshore. Even in Antarctic regions with in situ chlorophyll and Fe data, it is very challenging to deduce a significant link between Fe inputs and fertilization at a local scale because of the multiple factors that affect plankton growth on seasonal timescales (e.g. sediment also quenches light availability)- see for example the recent work by Forsch et al., which includes labile particle outflows (Forsch et al., 2021). To test a link between a specific cause and effect would require a statistically valid method (e.g. see prior work concerning icebergs (Wu and Hou, 2017)), perhaps the authors could consider consulting an oceanographer about this, but in this work it would be very difficult to do anything meaningful due to the long timescale. I'm not an expert on Antarctica, but perhaps finding areas of Antarctica with/without nunataks and testing whether there are statistically significant differences in how ocean chlorophyll dynamics behave in these regions might be insightful? This would have to be done in statistically valid way though. In any case, this would require a lot of work and would probably be a bit of a distraction from the core, new geochemical measurements presented.

In summary, I think the core geochemical measurements the authors have made are valuable and potentially useful in a number of contexts, but would probably be better suited to a short letter in a subject-specific journal without a briefer discussion of the relevance to other topics.

Specific comments by line number

16 Throughout, I don't think 'BioFe' is useful, the authors are testing for labile Fe phases. Bioavailability is a very complicated concept depending on multiple factors including lability and, in this case, particle dynamics. More generally, particles do not just affect pelagic primary production in the ocean and are, for example, major influences on near-shore benthic biogeochemical processes (e.g. see extensive work from Kogonsfjorden, Jørgensen et al., 2021). So why not just stick with 'labile' throughout?

28 "Delivery to free-floating icebergs, and coastal currents will deliver this BioFe to a larger reach of the Southern Ocean, strengthening the Southern Ocean biological pump" this concept is mentioned throughout the paper but is not well

developed. There is a lot of literature on the subject which discusses the complexity of sediment delivery, for example a relatively uncontroversial hypothesis substantiated by deposition patterns on the sea floor is that most sediment from ice shelves is deposited beneath the ice shelves (Smith et al., 2019). Accordingly, in some coastal sectors there's very little evidence of iron (particulate, or dissolved) emerging from the ice shelf edge (van Manen et al., 2022; Sedwick et al., 2022). Conversely in other areas, a 'trap and release' mechanism where glacier-derived particles are re-worked in sea ice or marine ice may drive Fe enrichment (Herraiz-Borreguero et al., 2016). As per my major comment, this story may be getting too far away from the core measurements but, if the authors want to discuss it or work in some more statistically valid chlorophyll analysis, I would encourage them to go a little deeper into the relevant literature concerning these processes.

Key words: suggest removing 'primary production' and 'nutrient delivery'

38 Sentence is too broad, iron-limitation is prevalent in some specific marine regions

45 There is a bit of a jump here between the narrative concerning Fe fluxes and their relevance to broad-scale processes and the details of this study. Just a comment concerning reader interest, but I felt like there was a missing paragraph to further explain the general concepts in the first paragraph which cover a lot e.g. if targeting a broad audience the less geological reader may want some of the basics about iron cycling in these environments, and the more geological reader may want some of the basics of the 'iron hypothesis' in the Southern Ocean (Martin and Fitzwater, 1988; Martínez-García et al., 2014)

58-74 This section could do with some references to substantiate these comments, are these well-established or just conceptual sediment routes/fates?

99-100 Somewhere it might be worth mentioning, given the marine context, how these compare to marine sediments. I'm not an expert on this, but if my unit conversions are correct the Fe leached throughout is actually at the very low end of marine sediment Fe liabilities e.g. see data from Svalbard where there are marine/cryosphere leached Fe measurements at quite good resolution in some locations (Laufer-Meiser et al., 2021)? This would make a big difference to the narrative as if glacier-derived sediment Fe liability is actually quite low compared to shelf sediments, then I would assume iceberg scouring of shelf sediments would potentially be a much more significant process in terms of making Fe available to the water column along coastal ice interfaces rather than the direct input of 'new' sediment released directly from ice?

132 "nutrient" Why not "Iron" or "Labile Fe"

146-151 This deduction is not statistically valid. There are alongside chlorophyll blooms in many (maybe most) regions worldwide, including regions which do not have glacier outflows, large iron inputs, or any evidence of plausible Fe-limitation. Light is a major limiting factor for phytoplankton growth in these environments, which is also strongly influenced by meltwater, so it's very difficult to prove much without some detailed statistical analysis that focus on a specific hypothesis. On its own, I don't think this really adds much to the text at present.

160-162 These seem to me quite big logic jumps, a warming climate will have a lot of effects, especially at 10-100 ka timescales. I suggest at least qualifying this with 'all other factors being constant'. A slight change in ice shelf dynamics for example, or sub ice-shelf circulation would presumably have large implications for sediment dispersal? Furthermore, these effects will be decoupled in time from the driver i.e. if the Fe source changes during warm climate periods, but then the resulting mineral products do not enter the ocean for 10-100 ka, they will not correspond to the same time period as they were sourced from, so this would not be a clear contributing factor to the glacier-interglacial iron hypothesis as proposed by Martin et al., rather it would be offset from the glacier-inter glacial fertilization scenario. I think the authors would perhaps need to refer to offshore sediment work to test these hypothesis, there are sediment cores in the Southern Ocean which record iceberg/glacier-derived sediment, the coverage may not be great, but these may be informative whether or not there is clear evidence for ice rafted debris changing on the spatial/timescales suggested if it is a broad-scale process as suggested.

170-172 This relates to ice shelf dynamics and may not be correct oceanographically (e.g. most ice shelf cavities where sediment enters the water column are pitch black and the residence time of marine water under neath ice shelves is often very long (up to 1-2 years) and probably often much longer than the residence time of particles in the water column

308-309 This is perhaps a bit broad, there is on-going discussion about this and critical issues do remain challenging to resolve, for example the standard NASA data products for chlorophyll a at the ice edge are not great, there are also particle interferences in areas with glacier plumes which are very difficult to remove. See for example recent work arguing for pronounced under-estimation of blooms in the Ross Sea (Chen et al., 2021).

324 This (very high!) temperature is definitely not an anomaly right? I assume this would have to be captured as a record-breaking weather anomaly so I assume should be verified by numerous weather station records? A sentence about this would be helpful.

Table 1. Is FeD : FeA actually a %? Or just a unit-less ratio. Maybe remind the reader what the abbreviations for the different types of iron mean

547 I am not sure this is strictly correct, my understanding was that the vast majority of ice melt (from ice shelves and icebergs) is subsurface as melt is driven by ocean heat rather than solar heating. Are there references to describe this scheme or is it conceptual? (Similar to an earlier comment when these processes were described in the text).

No comments on the Supplement file which all reads clearly.

References referred to

Chen, S., Smith Jr., W. O., and Yu, X.: Revisiting the Ocean Color Algorithms for Particulate Organic Carbon and Chlorophyll-a Concentrations in the Ross Sea, *J Geophys Res Oceans*, 126, e2021JC017749, <https://doi.org/https://doi.org/10.1029/2021JC017749>, 2021.

Forsch, K. O., Hahn-Woernle, L., Sherrell, R. M., Rocanova, V. J., Bu, K., Burdige, D., Vernet, M., and Barbeau, K. A.: Seasonal dispersal of fjord meltwaters as an important source of iron and manganese to coastal Antarctic phytoplankton, *Biogeosciences*, 18, 6349–6375, <https://doi.org/10.5194/bg-18-6349-2021>, 2021.

Herraiz-Borreguero, L., Lannuzel, D., van der Merwe, P., Treverrow, A., and Pedro, J. B.: Large flux of iron from the Amery Ice Shelf marine ice to Prydz Bay, East Antarctica, *J Geophys Res Oceans*, 121, 6009–6020, <https://doi.org/10.1002/2016JC011687>, 2016.

Jørgensen, B. B., Laufer, K., Michaud, A. B., and Wehrmann, L. M.: Biogeochemistry and microbiology of high Arctic marine

sediment ecosystems—Case study of Svalbard fjords, *Limnol Oceanogr*, 66, S273–S292, <https://doi.org/10.1002/lno.11551>, 2021.

Laufer-Meiser, K., Michaud, A. B., Maisch, M., Byrne, J. M., Kappler, A., Patterson, M. O., Røy, H., and Jørgensen, B. B.: Potentially bioavailable iron produced through benthic cycling in glaciated Arctic fjords of Svalbard, *Nat Commun*, 12, 1349, <https://doi.org/10.1038/s41467-021-21558-w>, 2021.

van Manen, M., Aoki, S., Brussaard, C. P. D., Conway, T. M., Eich, C., Gerringa, L. J. A., Jung, J., Kim, T.-W., Lee, S., Lee, Y., Reichart, G.-J., Tian, H.-A., Wille, F., and Middag, R.: The role of the Dotson Ice Shelf and Circumpolar Deep Water as driver and source of dissolved and particulate iron and manganese in the Amundsen Sea polynya, *Southern Ocean, Mar Chem*, 246, 104161, <https://doi.org/10.1016/j.marchem.2022.104161>, 2022.

Martin, J. H. and Fitzwater, S. E.: Iron-Deficiency Limits Phytoplankton Growth in the Northeast Pacific Subarctic, *Nature*, 331, 341–343, <https://doi.org/10.1038/331341a0>, 1988.

Martínez-García, A., Sigman, D. M., Ren, H., Anderson, R. F., Straub, M., Hodell, D. a, Jaccard, S. L., Eglinton, T. I., and Haug, G. H.: Iron fertilization of the Subantarctic ocean during the last ice age., *Science*, 343, 1347–1350, <https://doi.org/10.1126/science.1246848>, 2014.

Sedwick, P. N., Sohst, B. M., O'Hara, C., Stammerjohn, S. E., Loose, B., Dinniman, M. S., Buck, N. J., Resing, J. A., and Ackley, S. F.: Seasonal dynamics of dissolved iron on the Antarctic continental shelf: Late-fall observations from the Terra Nova Bay and Ross Ice Shelf polynyas, *J Geophys Res Oceans*, n/a, e2022JC018999, <https://doi.org/10.1029/2022JC018999>, 2022.

Smith, J. A., Graham, A. G. C., Post, A. L., Hillenbrand, C.-D., Bart, P. J., and Powell, R. D.: The marine geological imprint of Antarctic ice shelves, *Nat Commun*, 10, 5635, <https://doi.org/10.1038/s41467-019-13496-5>, 2019.

Wu, S.-Y. and Hou, S.: Impact of icebergs on net primary productivity in the Southern Ocean, *Cryosphere*, 11, 707–722, <https://doi.org/10.5194/tc-11-707-2017>, 2017.

Reviewer #2

(Remarks to the Author)

In the manuscript “thinning Antarctic glaciers expose high-altitude nunataks, delivering more bioavailable iron to the Southern Ocean”, the authors present measurements of ascorbate and dithionite extractable Fe from sediments collected from a glacier surface and nunataks in East Antarctica. These measurements are the first of their kind and provide valuable information about iron contents in continental material, which potentially gets transported to the ocean by glaciers. The study is concise and well written. I only have a few points that should be addressed by the authors in a revision:

As the word nunataks is probably not familiar to many readers, I suggest they shortly describe what they are.

Line 138: You estimate that it will take at least 10-100 ka for sediments sourced from the mountains to reach the ice front. I guess you should add some discussion on how the material could change in this time – it for sure will change. Right now it reads as if you assume the material is the same and will be transported like this to the ocean. But especially the more reactive “bioavailable” material will change.

Line 252-261: the concentrations of the chemicals should be given, as well as the extraction time and how they were extracted (shaking, dark?).

Reviewer #3

(Remarks to the Author)

It was my pleasure to review this manuscript from Dr. Winters, titled “Thinning Antarctic glaciers expose high-altitude nunataks, delivering more bioavailable iron to the Southern Ocean”. I do not have many comments as the manuscript is well-written and the science appears reasoned and sound.

My main criticism is with scale. This is a relatively small region in Antarctica with a relatively small number of samples. I commend the authors for not over-extrapolating their results, but I do feel that there is missing discussion on the broader context and interpretations of the results. As the EAIS thins, we anticipated a series of complicated responses and feedbacks. For example, warming conditions may actually decrease wind-blown sediment transport onto ice, assuming appreciable increases in soil water content. I believe a discussion on the representation of the region and broader significance giving warming scenarios is missing.

Line comments:

18: I recommend clarifying to state that the Southern Ocean is rich in major nutrients because Fe is an important micronutrient.

43: Please note that there have been studies focused on Fe delivery to the Southern Ocean from other areas in Antarctica. While BioFe was not quantified, this prior work is worth acknowledging. E.g., <https://doi.org/10.1029/2017JG004352>

66: I challenge the assertion that subglacial-derived BioFe uptake is limited. Subglacial plumes are common in many polar glacial systems. Since the plumes are buoyant, they can sometimes overcome diffusion to deliver nutrients to the surface (see work by Hopwood). These subglacial waters may be an extremely important source of nutrients to the photic zone.

115: I'm curious about these weathering processes on the nunataks. Rates of chemical weathering for Antarctic soils are very low without the sustained presence of liquid water. Did the authors estimate the degree of weathering or alteration for

the sediments?

133: Unclear what is meant by continental study. This is certainly not continent-wide.

151: Please see my earlier comment about subglacial plumes. Are the authors suggesting that the enhanced primary productivity is attributable to sediment-rich icebergs? It would have been nice to see a little more evidence for this, if that is indeed the argument.

187: Do these nutrients contribute more past the study area? I would have liked to read about considerations for contributions in warmer scenarios?

194: In warmer conditions, sediment transport via wind deflation should decrease due to increased soil cohesion. And while humidity and precipitation may increase, it's not likely enough for significant mass movement due to water flow. Do the authors expect more sediment delivery to the surrounding ice? I'm not sure thinning ice sheets will contribute more terrestrial sediments, especially compared to subglacial sources. There should probably be more discussion on this topic.

208: What are the snowfall rates? Isn't most snow sublimated in the winter and fall? How weathered are these sediments?

323: Diurnal fluctuations up to 25C seems quite high. Is this consistent with other work in Antarctica?

Fig 3: I'm not sure the thicker vs thinner ice sheet concept is properly captured. Is this why the pie charts are different sizes? I recommend making the condition distinctions clearer. Do the authors have relative magnitudes on BioFe delivery they can represent in this figure?

Again, this was an interesting article and my comments are relatively minor. This work will be an important contribution to geochemical studies in periglacial regions.

Version 1:

Reviewer comments:

Reviewer #1

(Remarks to the Author)

Thinning Antarctic glaciers expose high-altitude nunataks delivering more bioavailable iron to the Southern Ocean
I reviewed the original version of this submission. The revised version reads well and addressed most of the potential caveats I raised. The introduction particularly is much smoother, more suitable for a broad audience, and I found it easier to read. I have no further comments on the text, though the following very minor suggestions may improve accuracy or readability.

Very minor suggested edits

40 "with many originating" not sure this is grammatically correct, perhaps 'with many source terms'

72 perhaps 'resuspended or dissolved'

86 "is evidenced by the occurrence of large phytoplankton blooms" This is not strictly correct as any upwelling zone will always evidence a bloom during the growth season. Whether or not there is sufficient Fe supply to drawdown nutrients fully or alleviate Fe-limitation is a related but distinct question. Perhaps "The biological effect of these sub-glacial/basally-transported and englacial or supraglacial iron sources is challenging to constrain directly, but they may be a contributing iron source to large phytoplankton blooms that occur along the coastline".

108 Reference? Given this also depends very much on exactly when and where sediment is delivered suggest -unless there are specific references- "might be enriched"

125 "any other reported FeD data from the Antarctic continent." References?

132-140 Would it be helpful to quote a crustal average somewhere for reference?

179 This is not strictly correct as even if there was zero new Fe delivered from shelf or cryosphere sources, there would still be strong summertime blooms in all upwelling regions of the Southern Ocean. Perhaps "The potential for increased Fe delivery to enhance productivity in the ocean is supported by...." And/or an extra sentence something like "Whilst there are multiple iron sources to the coastal zone around Antarctica, which we cannot distinguish herein, macronutrient concentrations remain replete across practically the entire studied coastline during the growth season such that any increase in iron supply could support higher productivity"

183 Perhaps doesn't need changing, but just note the Gerringa et al., source actually got this a little wrong, the paper is widely cited as supporting the hypothesis as quoted here because it suggested a quite large glacier-derived Fe flux from direct measurements, and I think is the strongest study in the literature providing a clear link between melting Antarctic glaciers, increased Fe supply and Southern Ocean productivity. However, the flux calculation in the original paper was an order of magnitude too high due to a math error as per the correction on the article published later, which brings the finding much more in line with other work showing some effect but with difficulty fully distinguishing it from other processes

221 suggest 'to be directly useful'

Reviewer #3

(Remarks to the Author)

I appreciate the opportunity to review this revised manuscript by Winters et al. As I stated in my original review, I enjoyed learning about the author's work and felt the manuscript was strong. While some of the edits impacted the clarity of the manuscript, in general the authors have sufficiently, in my opinion, addressed the major comments from the original reviews. Below are just a handful of minor comments for consideration.

Introduction: To be completely honest, I overall preferred the original Introduction. The authors added a significant amount of information on under ice currents, but I think it detracts a bit from the story. In particular, the paragraph on that begins on line 68 with "Our simulated sediment transport" is a nice summary of recent literature, but shifts the focus of this entire section to currents underneath ice shelves, despite the novelty and importance of the near surface sediment dynamics originating from the nunatak erosion. While not essential, the authors may consider removing some of the emphasis on the currents to balance this section. Note that the case made in lines 62-63 regarding depth of sediment entrainment is important.

Lines 38-40: The grammar and structure of this sentence make it difficult to read/understand.

Lines 220-221 and 230-231: These read redundant.

Lines 248-251: The grammar and structure of this sentence make it difficult to read/understand.

Lines 378- 380: Are air and rock temperature data publicly available? I did not see a citation and did not see them included in the supplement.

Lines throughout: The original manuscript does a nice job of distinguishing BioFe, FeA, and FeD, and their relative importance for primary productivity. Oddly, I find the revisions to the terminology more confusing. The authors discuss the importance of BioFe in the introduction, but the term is essentially abandoned shortly after and in the discussion... it's a bit confusing.

The updates to Figure 3 are much clearer; the schematic is significantly improved. It's great!

Reviewer Comments

Reviewer #1 (Remarks to the Author):

Winter et al., present measurements of iron (Fe) lability from exposed mountain ridges in Antarctica. By modeling the route of sediment entrained in ice and carried to the coastline some time later they argue that the sediment has a fertilization effect increasing chlorophyll distribution in the adjacent Southern Ocean. The leaching techniques used to establish Fe lability are well established as useful and practical methods to roughly constrain the more-reactive Fe phases of most relevance to short-term biogeochemical cycles, rather than harder and more crystalline phases which are less reactive. The techniques as applied herein are consistent with prior work in other polar regions, which were largely pioneered by Raiswell et al.,. For obvious logistical reasons, a notably deficiency of prior work is that much of the sedimentary analysis to date has been conducted on samples from other glaciated regions in Greenland, Svalbard, and other lower latitude regions, with Antarctica poorly sampled by comparison. A strength of this new work is that it is based in an under sampled region.

Thanks for your thorough review, and your appreciation of our methods and approach to documenting samples in this important and under sampled region of the world.

The main weakness of the work is that the narrative and links to oceanographic processes are more challenging to address and I think these aspects of the work are quite weak and not so well supported as the core geochemistry. The main hypothesis is that labile Fe in sediment moving along ice flows from nunataks has a downstream fertilizing effect in the ocean, but -whilst plausible- demonstrating this with modern day data is difficult as presented.

As you later note, our research team has significant experience in geochemistry, glaciology and mountain processes. As such, and on your recommendation (also later on) we have worked with Dr Sian Henley from the University of Edinburgh to strengthen our understanding and presentation of sub-ice shelf and wider oceanographic processes in this re-submission. Dr Henley is now a co-author on the revised paper, and I'm sure, like us, you'll find the manuscript is much stronger in addressing oceanic processes as a result of her inputs.

First, the timescales are so long (10-100 ka) that there would be substantial de-coupling between any processes occurring at the nunataks and at the calving ice front or basal ice shelf at the coastline.

We agree that there could be substantial decoupling in time between weathering and delivery, and indeed this is shown by our particle tracking experiments, which are used as first order approximations of time (as we now note more explicitly in line 162, and in the caption for Figure 1). Provided long term storage is avoided, delivery will be inevitable, and rates of subaerially sourced sediment entering the ocean will increase as glaciers thin (and have thinned in this region since the Last Glacial Maximum). Whilst increased delivery time might allow more ferrihydrite conversion to less reactive minerals, it is equally possible that

it may enable the dissolution of less reactive phases and their conversion back to ferrihydrite. We have now added text resulting from this conversation in the first and second paragraph discussion section, so are grateful for you bringing this to our attention.

Second, the basic chlorophyll images used are not proof of a fertilizing effect e.g. chlorophyll is generally high in all coastal regions (globally and more specifically around Antarctica) irrespective of what nutrient/light/grazing limitation patterns are present in the coastal zones, and whether or not there are glacier/iron fluxes at the coastline. These levels of elevated chlorophyll, for obvious reasons, follow the prevailing currents which are likely alongshore. Even in Antarctic regions with in situ chlorophyll and Fe data, it is very challenging to deduce a significant link between Fe inputs and fertilization at a local scale because of the multiple factors that affect plankton growth on seasonal timescales (e.g. sediment also quenches light availability)- see for example the recent work by Forsch et al., which includes labile particle outflows (Forsch et al., 2021). To test a link between a specific cause and effect would require a statistically valid method (e.g. see prior work concerning icebergs (Wu and Hou, 2017)), perhaps the authors could consider consulting an oceanographer about this, but in this work it would be very difficult to do anything meaningful due to the long timescale. I'm not an expert on Antarctica, but perhaps finding areas of Antarctica with/without nunataks and testing whether there are statistically significant differences in how ocean chlorophyll dynamics behave in these regions might be insightful? This would have to be done in statistically valid way though. In any case, this would require a lot of work and would probably be a bit of a distraction from the core, new geochemical measurements presented.

We recognise our over-simplification of chlorophyll and ocean processes, which as you infer, was largely a result of our past research experience. We have taken on board your comments to include an oceanographer in our research team, and hope that you appreciate our combined efforts to significantly improve text related to chlorophyll concentrations and ocean processes throughout the manuscript, and references to it elsewhere (e.g. in figure captions). The annual-average chlorophyll map we showed in Figure 1 showed signals dampened by low autumn/winter values, so we now plot monthly climatology data for January (which is the month Arrigo et al. (2015) recorded peak net primary production in our study area, and maximum or near-maximum open water area).

In summary, I think the core geochemical measurements the authors have made are valuable and potentially useful in a number of contexts, but would probably be better suited to a short letter in a subject-specific journal without a briefer discussion of the relevance to other topics.

Thank you for highlighting the importance of our geochemical measurements. Now that we have placed them into a broader context (in both space and time) and made stronger links to the ocean, the manuscript is now much better placed for publication in Nature Communications. As you note, the work is “useful in a number of contexts” so we would welcome the opportunity for our work to be read and referenced by the broad readership of Nature Communications.

Specific comments by line number

16 Throughout, I don't think 'BioFe' is useful, the authors are testing for labile Fe phases. Bioavailability is a very complicated concept depending on multiple factors including lability and, in this case, particle dynamics. More generally, particles do not just affect pelagic primary production in the ocean and are, for example, major influences on near-shore benthic biogeochemical processes (e.g. see extensive work from Kognsfjorden, Jørgensen et al., 2021). So why not just stick with 'labile' throughout?

We have taken these comments on board and changed BioFe to labile Fe throughout the text, as appropriate.

28 "Delivery to free-floating icebergs, and coastal currents will deliver this BioFe to a larger reach of the Southern Ocean, strengthening the Southern Ocean biological pump" this concept is mentioned throughout the paper but is not well developed. There is a lot of literature on the subject which discusses the complexity of sediment delivery, for example a relatively uncontroversial hypothesis substantiated by deposition patterns on the sea floor is that most sediment from ice shelves is deposited beneath the ice shelves (Smith et al., 2019). Accordingly, in some coastal sectors there's very little evidence of iron (particulate, or dissolved) emerging from the ice shelf edge (van Manen et al., 2022; Sedwick et al., 2022). Conversely in other areas, a 'trap and release' mechanism where glacier-derived particles are re-worked in sea ice or marine ice may drive Fe enrichment (Herraiz-Borreguero et al., 2016). As per my major comment, this story may be getting too far away from the core measurements but, if the authors want to discuss it or work in some more statistically valid chlorophyll analysis, I would encourage them to go a little deeper into the relevant literature concerning these processes.

Thank you for your comments and reference suggestions – they have been very helpful. Our new author, Dr Sian Henley has helped us to substantially improve our manuscript in this regard – weaving in discussions related to your text throughout (see in particular - lines 68-90 and 219-22) – where we have also added in a variety of new supporting references. We acknowledge our original over-simplification and hope that you find the manuscript much more compelling in these aspects now.

Key words: suggest removing 'primary production' and 'nutrient delivery'

We have kept these words in, as we feel our work speaks directly to "nutrient delivery", and the term "primary production" will help researchers to find our work and relate our findings to theirs in an appropriate way.

*38 Sentence is too broad, iron-limitation is prevalent in some specific marine regions
In light of this comment, and the comment below (for line 45), we have created a new introductory paragraph (line 68-90) that outlines general concepts in much more detail.*

45 There is a bit of a jump here between the narrative concerning Fe fluxes and their relevance to broad-scale processes and the details of this study. Just a comment

concerning reader interest, but I felt like there was a missing paragraph to further explain the general concepts in the first paragraph which cover a lot e.g. if targeting a broad audience the less geological reader may want some of the basics about iron cycling in these environments, and the more geological reader may want some of the basics of the ‘iron hypothesis’ in the Southern Ocean (Martin and Fitzwater, 1988; Martínez-García et al., 2014)

Thanks for your suggestion. We have added a new introductory paragraph as suggested (lines 68-90) and feel that the manuscript has been significantly improved as a result.

58-74 *This section could do with some references to substantiate these comments, are these well-established or just conceptual sediment routes/fates?*

These are observed mechanisms/processes of subaerial sediment production from Antarctica or other cold regions. We have amended the text (now in section that begins on line 68) to improve clarity and added supporting references as you suggest.

99-100 *Somewhere it might be worth mentioning, given the marine context, how these compare to marine sediments. I’m not an expert on this, but if my unit conversions are correct the Fe leached throughout is actually at the very low end of marine sediment Fe labilities e.g. see data from Svalbard where there are marine/cryosphere leached Fe measurements at quite good resolution in some locations (Laufer-Meiser et al., 2021)? This would make a big difference to the narrative as if glacier-derived sediment Fe lability is actually quite low compared to shelf sediments, then I would assume iceberg scouring of shelf sediments would potentially be a much more significant process in terms of making Fe available to the water column along coastal ice interfaces rather than the direct input of ‘new’ sediment released directly from ice?*

We have reviewed the Laufer-Meiser et al. 2021 paper and see that they report much higher values, but they come from red-bed sedimentary rocks which means they are not directly comparable to our study site. We have however compared our results to those from icebergs and melting ice (lines 117-1188) as well as continental sediments from West Antarctica (line 126-128). We have also added in new discussions about the importance of sediment/ice/water interactions past the grounding line throughout the manuscript (see in particular lines 219-223 and 228-240).

132 *“nutrient” Why not “Iron” or “Labile Fe”*

We have removed this heading (due to formatting guidelines) so a change is no longer required.

146-151 *This deduction is not statistically valid. There are alongside chlorophyll blooms in many (maybe most) regions worldwide, including regions which do not have glacier outflows, large iron inputs, or any evidence of plausible Fe-limitation. Light is a major limiting factor for phytoplankton growth in these environments, which is also strongly influenced by meltwater, so it’s very difficult to prove much without some detailed statistical analysis that focus on a specific hypothesis. On its own, I don’t think this really adds much to the text at present.*

We have added in new text that directly references the importance of light, meltwater and ocean/sediment interactions throughout the manuscript and we have added new text into this section (now beginning on line 179) to place our discussions into a wider context, with references that speak directly to phytoplankton biomass and associated iron supply research in Dronning Maud Land.

160-162 These seem to me quite big logic jumps, a warming climate will have a lot of effects, especially at 10-100 ka timescales. I suggest at least qualifying this with 'all other factors being constant'. A slight change in ice shelf dynamics for example, or sub ice-shelf circulation would presumably have large implications for sediment dispersal? Furthermore, these effects will be decoupled in time from the driver i.e. if the Fe source changes during warm climate periods, but then the resulting mineral products do not enter the ocean for 10-100 ka, they will not correspond to the same time period as they were sourced from, so this would not be a clear contributing factor to the glacier-interglacial iron hypothesis as proposed by Martin et al., rather it would be offset from the glacier-inter glacial fertilization scenario. I think the authors would perhaps need to refer to offshore sediment work to test these hypothesis, there are sediment cores in the Southern Ocean which record iceberg/glacier-derived sediment, the coverage may not be great, but these may be informative whether or not there is clear evidence for ice rafted debris changing on the spatial/timescales suggested if it is a broad-scale process as suggested.

Hopefully you will find that our more detailed, revised manuscript makes clearer links between steps in our logic and provides a more coherent narrative overall. On this specific point, we now note in line 36 that “over glacial-interglacial cycles, iron supply to the Southern Ocean from Antarctica has varied significantly and has driven major changes in primary production and carbon export” (with references) and have provided more discussion on lag times and the impact these may have on lines 164-166 and later, in lines 204-209. We have also added in the qualifying statement you suggest “with all other factors being consistent” in line 201. We have not examined sediment core records in detail, because this is beyond the scope of the study, but we have compared the timescales of changes discussed in our study to timescales of other iron delivery mechanisms to the Southern Ocean.

170-172 This relates to ice shelf dynamics and may not be correct oceanographically (e.g. most ice shelf cavities where sediment enters the water column are pitch black and the residence time of marine water under neath ice shelves is often very long (up to 1-2 years) and probably often much longer than the residence time of particles in the water column
We acknowledge the over-simplified text we presented in lines 170-172 of the original manuscript. We now explore sediment/ice/water interactions past the grounding line in much more detail in lines 218-223 (after outlining these important concepts in the introduction (mainly lines 68-90).

308-309 This is perhaps a bit broad, there is on-going discussion about this and critical issues do remain challenging to resolve, for example the standard NASA data products for

chlorophyll a at the ice edge are not great, there are also particle interferences in areas with glacier plumes which are very difficult to remove. See for example recent work arguing for pronounced under-estimation of blooms in the Ross Sea (Chen et al., 2021).

Thanks for highlighting these discussions. The annual-average chlorophyll map we originally showed in Figure 1a included signals dampened by low autumn/winter values, so we now plot monthly climatology data for January in Figure 1a instead - which is the month Arrigo et al. (2015) recorded peak net primary production in our study area, and maximum or near-maximum open water area. As there are some concerns over the accuracy of chlorophyll concentrations from satellite-derived ocean colour measurements in high-latitude waters, we focus on relative concentrations rather than absolute values in the paper, and now mention this in lines 366-367 and 370-371 of our manuscript. Another concern, that satellites cannot detect chlorophyll below the upper optical depth of the ocean (and therefore cannot measure chlorophyll at depth), is partially alleviated by the very high correlation between surface chlorophyll and depth-integrated chlorophyll within the mixed layer of the Southern Ocean (Arrigo et al., 1998). As a result, the concentrations of chlorophyll presented in Figure 1a are representative of total concentrations at the temporal and spatial scales we examine. We have added this text to the 'Chlorophyll extractions' paragraph, beginning on line 362.

324 This (very high!) temperature is definitely not an anomaly right? I assume this would have to be captured as a record-breaking weather anomaly so I assume should be verified by numerous weather station records? A sentence about this would be helpful.

No, this is not an anomaly, and it doesn't represent 'weather'. We apologise if the text in our manuscript was confusing. We present air temperature measurements from a local weather station as well as temperature recordings directly from a local bedrock site. We placed 3 ultra-low temperature data loggers (which were purchased new for the project and each delivered with a ISO17025 accredited calibration certificate) at 3 different angles on bedrock (which happened to be dark in colour, like a lot of the nunataks we explored) and find that our results are compatible and expected - with north facing slopes warming more than south facing slopes.

There are not many studies that present rock temperature data in Antarctica, but Lamp et al. (2016) record temperatures up to 28°C on detaching flakes on surface clasts in the Dry Valleys (when ambient air temperature was -15°C) and rock temperatures of almost 20°C were recorded by Hall and André (2001) in the Antarctic Peninsula. We have referenced Lamp et al. (2016) (see line 393) in our revised manuscript and amended the text in this section (beginning on line 386) to try to make our findings clearer.

Table 1. Is FeD : FeA actually a %? Or just a unit-less ratio. Maybe remind the reader what the abbreviations for the different types of iron mean

Yes, this is a unit-less ratio and we have amended the table and related manuscript text (mainly in lines 110-130) to make this clear. We have also updated Table 1 with abbreviation definitions, as you suggest.

547 I am not sure this is strictly correct, my understanding was that the vast majority of ice melt (from ice shelves and icebergs) is subsurface as melt is driven by ocean heat rather than solar heating. Are there references to describe this scheme or is it conceptual? (Similar to an earlier comment when these processes were described in the text).

Apologies for any confusion. Like you, we understand that the vast majority of ice melt comes from the underside of the ice and we have tried to clarify this throughout the manuscript. We hope that you agree that our new introductory paragraph (lines 68-90) really helps to outline processes beyond the grounding line in more detail. We have also revised Figure 3 to make our hypotheses clearer (and changed the caption as a result).

No comments on the Supplement file which all reads clearly.

Super! Thanks! We really appreciate all the time that went into your review, and hope that you find the manuscript much stronger as a result of these discussions.

References referred to

- Chen, S., Smith Jr., W. O., and Yu, X.: Revisiting the Ocean Color Algorithms for Particulate Organic Carbon and Chlorophyll-*a* Concentrations in the Ross Sea, *J Geophys Res Oceans*, 126, e2021JC017749, <https://doi.org/10.1029/2021JC017749>, 2021.
- Forsch, K. O., Hahn-Woernle, L., Sherrell, R. M., Roccanova, V. J., Bu, K., Burdige, D., Vernet, M., and Barbeau, K. A.: Seasonal dispersal of fjord meltwaters as an important source of iron and manganese to coastal Antarctic phytoplankton, *Biogeosciences*, 18, 6349–6375, <https://doi.org/10.5194/bg-18-6349-2021>, 2021.
- Herraiz-Borreguero, L., Lannuzel, D., van der Merwe, P., Treverrow, A., and Pedro, J. B.: Large flux of iron from the Amery Ice Shelf marine ice to Prydz Bay, East Antarctica, *J Geophys Res Oceans*, 121, 6009–6020, <https://doi.org/10.1002/2016JC011687>, 2016.
- Jørgensen, B. B., Laufer, K., Michaud, A. B., and Wehrmann, L. M.: Biogeochemistry and microbiology of high Arctic marine sediment ecosystems—Case study of Svalbard fjords, *Limnol Oceanogr*, 66, S273–S292, <https://doi.org/10.1002/lno.11551>, 2021.
- Laufer-Meiser, K., Michaud, A. B., Maisch, M., Byrne, J. M., Kappler, A., Patterson, M. O., Røy, H., and Jørgensen, B. B.: Potentially bioavailable iron produced through benthic cycling in glaciated Arctic fjords of Svalbard, *Nat Commun*, 12, 1349, <https://doi.org/10.1038/s41467-021-21558-w>, 2021.
- van Manen, M., Aoki, S., Brussaard, C. P. D., Conway, T. M., Eich, C., Gerringa, L. J. A., Jung, J., Kim, T.-W., Lee, S., Lee, Y., Reichart, G.-J., Tian, H.-A., Wille, F., and Middag, R.: The role of the Dotson Ice Shelf and Circumpolar Deep Water as driver and source of dissolved and particulate iron and manganese in the Amundsen Sea polynya, Southern Ocean, *Mar Chem*, 246, 104161, <https://doi.org/10.1016/j.marchem.2022.104161>, 2022.
- Martin, J. H. and Fitzwater, S. E.: Iron-Deficiency Limits Phytoplankton Growth in the Northeast Pacific Subarctic, *Nature*, 331, 341–343, <https://doi.org/10.1038/331341a0>, 1988.
- Martínez-García, A., Sigman, D. M., Ren, H., Anderson, R. F., Straub, M., Hodell, D. a., Jaccard, S. L., Eglinton, T. I., and Haug, G. H.: Iron fertilization of the Subantarctic ocean during the last ice age., *Science*, 343, 1347–1350, <https://doi.org/10.1126/science.1246848>,

2014.

Sedwick, P. N., Sohst, B. M., O'Hara, C., Stammerjohn, S. E., Loose, B., Dinniman, M. S., Buck, N. J., Resing, J. A., and Ackley, S. F.: Seasonal dynamics of dissolved iron on the Antarctic continental shelf: Late-fall observations from the Terra Nova Bay and Ross Ice Shelf polynyas, *J Geophys Res Oceans*, n/a, e2022JC018999, <https://doi.org/10.1029/2022JC018999>, 2022.

Smith, J. A., Graham, A. G. C., Post, A. L., Hillenbrand, C.-D., Bart, P. J., and Powell, R. D.: The marine geological imprint of Antarctic ice shelves, *Nat Commun*, 10, 5635, <https://doi.org/10.1038/s41467-019-13496-5>, 2019.

Wu, S.-Y. and Hou, S.: Impact of icebergs on net primary productivity in the Southern Ocean, *Cryosphere*, 11, 707–722, <https://doi.org/10.5194/tc-11-707-2017>, 2017.

Thanks! These are very helpful.

Reviewer #2 (Remarks to the Author):

In the manuscript “thinning Antarctic glaciers expose high-altitude nunataks, delivering more bioavailable iron to the Southern Ocean”, the authors present measurements of ascorbate and dithionite extractable Fe from sediments collected from a glacier surface and nunataks in East Antarctica. These measurements are the first of their kind and provide valuable information about iron contents in continental material, which potentially gets transported to the ocean by glaciers. The study is concise and well written. I only have a few points that should be addressed by the authors in a revision:

Thanks for reviewing our manuscript and highlighting the importance of our work. We are glad that you find the study concise and well-written.

As the word nunataks is probably not familiar to many readers, I suggest they shortly describe what they are.

We have now defined the word nunatak the first time it appears in the text (line 55) and have carefully considered the use of the word throughout the manuscript.

Line 138: You estimate that it will take at least 10-100 ka for sediments sourced from the mountains to reach the ice front. I guess you should add some discussion on how the material could change in this time – it for sure will change. Right now it reads as if you assume the material is the same and will be transported like this to the ocean. But especially the more reactive “bioavailable” material will change.

Agreed, but the change may not be only towards less reactive phases, as dissolution and reprecipitation may occur as re-freezing concentrates Fe, and there may be particle breakdown during these processes. We have now added the following text to lines 166-129 as a result of these discussions – “It is hard to quantify how material will change during transport, but we expect that dissolution and reprecipitation may occur as re-freezing concentrates iron, and that there may be particle breakdown during these processes, which may act to increase iron bioavailability.”

Line 252-261: the concentrations of the chemicals should be given, as well as the extraction time and how they were extracted (shaking, dark?).

Agreed. These details have now been added (see lines 310-320).

Reviewer #3 (Remarks to the Author):

It was my pleasure to review this manuscript from Dr. Winters, titled “Thinning Antarctic glaciers expose high-altitude nunataks, delivering more bioavailable iron to the Southern Ocean”. I do not have many comments as the manuscript is well-written and the science appears reasoned and sound.

Thank you for reviewing our paper and highlighting our efforts to write a clear and concise paper.

My main criticism is with scale. This is a relatively small region in Antarctica with a relatively small number of samples. I commend the authors for not over-extrapolating their results, but I do feel that there is missing discussion on the broader context and interpretations of the results. As the EAIS thins, we anticipated a series of complicated responses and feedbacks. For example, warming conditions may actually decrease wind-blown sediment transport onto ice, assuming appreciable increases in soil water content. I believe a discussion on the representation of the region and broader significance giving warming scenarios is missing.

Thank you for noticing the care we have taken to not over-extrapolate our results. Whilst we have revised the manuscript, in accordance with your suggestions, and that of the Editor and other reviewers, we have tried to be clear where we present our results, and how representative they may be of the region and more broadly. We have now added some broader context, highlighting how our work fits into longer-term processes of glacial/interglacial sediment transfer (line 36) and noted where our findings might fit in or help support research in similar high-latitude regions on Earth, as well as the icy surface of Mars (lines 197-199) – to make our paper more useful to the community, supporting the broad readership of Nature Communications.

Line comments:

18: I recommend clarifying to state that the Southern Ocean is rich in major nutrients because Fe is an important micronutrient.

We have changed the abstract to clarify the importance of iron as a micronutrient. We have not mentioned major nutrients here, to keep the abstract as succinct and focused as possible. We have also checked our use of the word nutrient throughout and clarified where needed. Thanks for bringing this to our attention.

43: Please note that there have been studies focused on Fe delivery to the Southern Ocean from other areas in Antarctica. While BioFe was not quantified, this prior work is worth acknowledging. E.g., <https://doi.org/10.1029/2017JG004352>

The introductory paragraph is now more encompassing, referencing a wider variety of material, including this recommended paper.

66: I challenge the assertion that subglacial-derived BioFe uptake is limited. Subglacial plumes are common in many polar glacial systems. Since the plumes are buoyant, they can sometimes overcome diffusion to deliver nutrients to the surface (see work by Hopwood).

These subglacial waters may be an extremely important source of nutrients to the photic zone.

Thank you for bringing this to our attention. Our new co-author, Dr Sian Henley has added substantial information about water/ice/sediment interactions from the grounding line to the calving margin (and beyond) throughout the manuscript; introducing these important systems in the third paragraph of the manuscript (lines 68-90) and outlining how they impact our study site and findings in the discussion section (particularly in lines 179-191, 219-223, 229-240).

115: I'm curious about these weathering processes on the nunataks. Rates of chemical weathering for Antarctic soils are very low without the sustained presence of liquid water. Did the authors estimate the degree of weathering or alteration for the sediments?

No we didn't but we agree that this would be great to know. The observed weathering is confined to isolated particles and rock surfaces so it would be difficult to estimate meaningful degrees of weathering. It is equally difficult to quantify the masses of particles as related to their weathered source (how much weathered material from how much nunatak rock). We note that chemical analysis of surface phases in isolation from their host rock might help, but again there is a mass issue; how much weathered material has been lost.

133: Unclear what is meant by continental study. This is certainly not continent-wide.

We meant land-based rather than ocean based, but agree that this is not obvious, so we have changed the sentence accordingly in lines 156-166, which now reads "sediments across our glaciated mountain study site".

151: Please see my earlier comment about subglacial plumes. Are the authors suggesting that the enhanced primary productivity is attributable to sediment-rich icebergs? It would have been nice to see a little more evidence for this, if that is indeed the argument.

We've added a new paragraph to the discussion (starting at line 179) to describe variations in phytoplankton biomass in the ocean adjacent to Dronning Maud Land and the potential drivers of these variations in more detail (with new references included).

187: Do these nutrients contribute more past the study area? I would have liked to read about considerations for contributions in warmer scenarios?

We have amended text in the discussion to make these statements more clear. Lines 197-190 now states that Chlorophyll a concentrations for January reveal phytoplankton blooms along the coastline, extending up to 175 km from the ice margin.

The text in the paragraph beginning on line 193 now outlines how sediment supply may increase in the future as temperatures rise and ice sheets thin.

We have also substantially revised figure 3 to more explicitly outline the changes we expect to see in warmer scenarios (schematic b). We describe this figure in lines 225-240.

194: In warmer conditions, sediment transport via wind deflation should decrease due to

increased soil cohesion. And while humidity and precipitation may increase, it's not likely enough for significant mass movement due to water flow. Do the authors expect more sediment delivery to the surrounding ice? I'm not sure thinning ice sheets will contribute more terrestrial sediments, especially compared to subglacial sources. There should probably be more discussion on this topic.

Warmer conditions are likely to increase the frequency and magnitude of mass movements from nunataks, and subsequent sediment delivery to ice. This is what we are recording in other cold regions of the world (which we now reference in line 247). So yes, we expect more terrestrial sediment delivery to ice. Increased humidity and precipitation condition and prepare rock-failure zones along with what has been commonly termed 'debuitressing' which encompasses temperature/water changes but also rock-mass damage during glacial/deglacial cycles. It is now well established that subaerial delivery via mass movements increases rapidly as ice thins/retreats, from fresh bedrock failures and from the delivery of metastable sediment stores such as moraines (Westoby et al., 2016). We have added more text and references to support our statements in this paragraph (lines 242-254).

The question about proportional contribution of subaerially vs. subglacially sourced sediment during warmer conditions is an interesting one. It is very difficult to define the proportion of subglacial material in Antarctica, and whether the generation and flux of sediment from this part of the glacier system will increase or decrease in a warming climate (it is probably going to be highly variable). The point we are trying to make is that sediment has been transported through glacial systems in Antarctica over glacial-interglacial cycles (line 36-38) and that the pathway to the ocean is key – as it's the subaerial material that will reach the ocean more directly via calving and melting at the ice shelf front – and could be more effective at priming productivity (due to increased FeD – see results). We have tried to make this clearer in our revised Figure 3, and associated text (lines 225-240).

208: What are the snowfall rates? Isn't most snow sublimated in the winter and fall? How weathered are these sediments?

We do not have an accurate measure of snowfall rates in this area, but our past work on wind-blown snow dispersal in Antarctic mountains (e.g. Mills *et al.*, 2019, *Journal of Glaciology*) highlights the importance of blowing snow in mountain regions, and having spent quite a lot of time at our study site, it is clear to us that snow often blows in and quickly sublimates away in the austral summer, when rock temperatures are high as a result of solar gain and thermal mass.

323: Diurnal fluctuations up to 25C seems quite high. Is this consistent with other work in Antarctica?

Reviewer 1 raised a similar question. We responded there noting that the CryoTemp instrument we deploy is an ultra-low temperature data logger capable of recording temperatures as low as -86 °C (-122.8 °F). We purchased three of these new for the project, and they were each delivered with an ISO17025 accredited calibration certificate. We

placed sensors at 3 different angles, and results are compatible, and expected (with north facing slopes warming more than south facing slopes). As a result, we are confident that the measurements we present are correct. The rocks did feel very warm to touch at mid-day!

There are not many studies that present rock temperature data in Antarctica, but Lamp et al. (2016) record temperatures up to 28°C on detaching flakes on surface clasts in the Dry Valleys (when ambient air temperature was -15°C) and rock temperatures of almost 20°C were recorded by Hall and André (2001) in the Antarctic Peninsula. We have now referenced Lamp et al. (2016) in our revised manuscript (line 393) and amended the text in lines 386-402 to try to make our results clearer (and more relatable to other studies).

Fig 3: I'm not sure the thicker vs thinner ice sheet concept is properly captured. Is this why the pie charts are different sizes? I recommend making the condition distinctions clearer. Do the authors have relative magnitudes on BioFe delivery they can represent in this figure? We have revised figure 3 to better capture the thicker vs thinner ice sheet and the impact this has on sediment transport routes to the ocean.

Again, this was an interesting article and my comments are relatively minor. This work will be an important contribution to geochemical studies in periglacial regions.

Thank you for your kind words, and for summarising your review highlighting the importance of our work. We hope you like the revised, more compelling paper!

Thank you for taking the time to review our manuscript “Thinning Antarctic glaciers expose high-altitude nunataks delivering more bioavailable iron to the Southern Ocean”. We have addressed all reviewer comments below, where our commentary is coloured blue, and we refer to line numbers in the final manuscript.

Reviewer #1 (Remarks to the Author):

I reviewed the original version of this submission. The revised version reads well and addressed most of the potential caveats I raised. The introduction particularly is much smoother, more suitable for a broad audience, and I found it easier to read. I have no further comments on the text, though the following very minor suggestions may improve accuracy or readability.

Thank you for your positive statements and for the time you have taken to review our work.

Very minor suggested edits

40 “with many originating” not sure this is grammatically correct, perhaps ‘with many source terms’

This text now says “with many of these source terms originating from the Antarctic continent” (line 36).

72 perhaps ‘resuspended or dissolved’

We have added in “or dissolved” on line 69.

86 “is evidenced by the occurrence of large phytoplankton blooms” This is not strictly correct as any upwelling zone will always evidence a bloom during the growth season. Whether or not there is sufficient Fe supply to drawdown nutrients fully or alleviate Fe-limitation is a related but distinct question. Perhaps “The biological effect of these subglacial/basally-transported and englacial or supraglacial iron sources is challenging to constrain directly, but they may be a contributing iron source to large phytoplankton blooms that occur along the coastline”.

We have revised this sentence in light of your comments. This sentence now reads “The biological effect of these subglacial/basally-transported and englacial or supraglacial iron sources is challenging to constrain directly, but they are likely to contribute iron supply to the large phytoplankton blooms that occur along the coastline⁴⁰” (lines 80-82).

108 Reference? Given this also depends very much on exactly when and where sediment is delivered suggest -unless there are specific references- “might be enriched” We already referenced the work of Shaked and Lis (2012) in this sentence, but have reworded the sentence a little to make this reference more clearly linked. The sentence now reads “Whilst we note that at the point of delivery to seawater, ferrihydrite (measured as FeA) is more labile than FeD, previous study has shown that FeD may become more bioavailable after delivery to seawater, due to processes that include the in-situ recycling of cellular iron through grazing and viral lysis⁴⁸” (line 101-104).

125 “any other reported FeD data from the Antarctic continent.” References?

Thanks for picking up this unintended omission. We have now referenced Raiswell et al., (2016) here (line 122).

132-140 Would it be helpful to quote a crustal average somewhere for reference?

We have considered this request but don't feel it is required.

179 This is not strictly correct as even if there was zero new Fe delivered from shelf or cryosphere sources, there would still be strong summertime blooms in all upwelling regions of the Southern Ocean. Perhaps “The potential for increased Fe delivery to enhance productivity in the ocean is supported by...” And/or an extra sentence something like “Whilst there are multiple iron sources to the coastal zone around Antarctica, which we cannot distinguish herein, macronutrient concentrations remain replete across practically the entire studied coastline during the growth season such that any increase in iron supply could support higher productivity”

We have amended this sentence accordingly. It now reads “The potential for iron delivery to enhance productivity in the ocean is supported by” – as you suggest (line 176). We have not mentioned macronutrients so feel the second suggestion is unnecessary, as it will not support the main arguments we put forward in the paper.

183 Perhaps doesn't need changing, but just note the Gerringa et al., source actually got this a little wrong, the paper is widely cited as supporting the hypothesis as quoted here because it suggested a quite large glacier-derived Fe flux from direct measurements, and I think is the strongest study in the literature providing a clear link between melting Antarctic glaciers, increased Fe supply and Southern Ocean productivity. However, the flux calculation in the original paper was an order of magnitude too high due to a math error as per the correction on the article published later, which brings the finding much more in line with other work showing some effect but with difficulty fully distinguishing it from other processes

Yes, we noticed that corrigendum too, but as you say – the reference is still valid, so we haven't made any changes to our manuscript.

221 suggest ‘to be directly useful’

We have made this suggested change (line 219).

Reviewer #3 (Remarks to the Author):

I appreciate the opportunity to review this revised manuscript by Winters et al. As I stated in my original review, I enjoyed learning about the author's work and felt the manuscript was strong. While some of the edits impacted the clarity of the manuscript, in

general the authors have sufficiently, in my opinion, addressed the major comments from the original reviews. Below are just a handful of minor comments for consideration.
Thank you for reviewing our manuscript once more and for being so open in your review.

Introduction: To be completely honest, I overall preferred the original Introduction. The authors added a significant amount of information on under ice currents, but I think it detracts a bit from the story. In particular, the paragraph on that begins on line 68 with "Our simulated sediment transport" is a nice summary of recent literature, but shifts the focus of this entire section to currents underneath ice shelves, despite the novelty and importance of the near surface sediment dynamics originating from the nunatak erosion. While not essential, the authors may consider removing some of the emphasis on the currents to balance this section. Note that the case made in lines 62-63 regarding depth of sediment entrainment is important.

We have reviewed the paragraph that begins "Our Simulated sediment transport routes" and made a few changes to streamline the text. We have kept the statements about depth of sediment entrainment, as we also agree this is very important.

Lines 38-40: The grammar and structure of this sentence make it difficult to read/understand.

Reviewer 1 pointed out something similar. This sentence now reads "Iron supply to the Southern Ocean has been documented via atmospheric, oceanic, cryogenic and terrestrial mechanisms (e.g. Gerringa et al., 2012⁶; Henley et al., 2020⁷; Tagliabue et al., 2017⁸), with many of these source terms originating from the Antarctic continent and continental shelves^{9,10,11}" (lines 34-37).

Lines 220-221 and 230-231: These read redundant.

We were asked to add this text in the first round of reviews and in the text you reviewed we specifically mention the word "useful", which is the opposite of redundant. We have reviewed this carefully and have decided to keep the text relatively unchanged (except for the addition of the word 'directly' on line 219) – as this helps us maintain a consistent narrative through the paper.

Lines 248-251: The grammar and structure of this sentence make it difficult to read/understand.

We have now re-worded the text. It now reads "We hypothesise that these factors will increase sediment (and iron) flux from subaerial sources. High-level glacial transport will help to deliver these newly exposed/released nunatak sediments beyond the grounding line to calving margins, where meltwater and free-floating icebergs can release sediment-derived nutrients over large areas of the coastal ocean, where they can be utilised by primary producers" (lines 245-249).

Lines 378- 380: Are air and rock temperature data publicly available? I did not see a citation and did not see them included in the supplement.

This is a very good suggestion. We have included the data values in a new data supplement as you suggest.

Lines throughout: The original manuscript does a nice job of distinguishing BioFe, FeA, and FeD, and their relative importance for primary productivity. Oddly, I find the revisions to the terminology more confusing. The authors discuss the importance of BioFe in the introduction, but the term is essentially abandoned shortly after and in the discussion... it's a bit confusing.

Thanks for raising this. We have reviewed the manuscript and have tried to streamline the terminology. We have deleted the word labile in line 45, 84 and 85.

The updates to Figure 3 are much clearer; the schematic is significantly improved. It's great!

Thanks! We love it too!